# Effects of Apple Vinegar, Mouthwashes, and Bleaching on Color Stability and Surface Properties of Fiber-Reinforced and Non-Reinforced Restorative Materials

**DOI:** 10.3390/polym17182552

**Published:** 2025-09-21

**Authors:** Kerem Yılmaz, Tuğçe Odabaş Hajiyev, Gökçe Özcan Altınsoy, Mehmet Mustafa Özarslan

**Affiliations:** 1Department of Prosthodontics, Faculty of Dentistry, Antalya Bilim University, 07190 Antalya, Turkey; 2Bilimdent Oral and Dental Health Center, Antalya Bilim University, 07190 Antalya, Turkey; 3Department of Restorative Dentistry, Faculty of Dentistry, Antalya Bilim University, 07190 Antalya, Turkey; 4Department of Pediatric Dentistry, Faculty of Dentistry, Antalya Bilim University, 07190 Antalya, Turkey

**Keywords:** color, fiber reinforced composites, hydrogen peroxide, mouthwashes, nanocomposites, surface roughness

## Abstract

The aim of this study was to investigate the effects of apple cider vinegar (ACV), various mouthwashes and bleaching on the color and surface roughness of fiber strip-reinforced and unreinforced restorative materials. The materials were resin composite (RC), resin-nanoceramic (RNC), and polymer-infiltrated ceramic network (PICN); the mouthwashes were chlorhexidine with alcohol (CXA), chlorhexidine without alcohol (CX), herbal with alcohol (HRA), and herbal without alcohol (HR). Measurements were performed at T0 (baseline), T1 (1 day), T2 (2.5 days) and T3 (after bleaching). Analysis of variance (ANOVA) and Bonferroni analyses revealed that roughness from T0–T3 was highest for RNC and lowest for PICN. Regarding the solutions, the highest increase was in ACV and lowest in artificial saliva (*p* < 0.001). At T0–T2, color change (ΔE_00_) and whiteness index change (ΔWI_D_) were highest in CXA and lowest in HR. At T2–T3, ΔE_00_ was highest in ACV, while ΔWI_D_ was highest in CXA (*p* < 0.001). Although the roughness exceeded the bacterial adhesion threshold, the effect of bleaching was not considerable. Color and whiteness changes generally did not exceed the acceptability threshold. Fiber strip position did not affect roughness. However, a strip in the middle layer had higher impact on color and whiteness than the one in the top layer.

## 1. Introduction

Throughout history, resin composites (RCs) have been widely preferred materials for mimicking the natural appearance of teeth [1]. The first resins used for dental fillings were based on polymethyl methacrylate (PMMA), which exhibited polymerization shrinkage up to 6% and insufficient wear resistance. In 1962, to overcome these disadvantages, Bowen developed bisphenol A glycidyl methacrylate (Bis-GMA). Bis-GMA is a high-molecular-weight, low-viscosity dimethacrylate that exhibits lower polymerization shrinkage and, owing to its aromatic structure, forms stiffer and more durable polymers [2]. Subsequently, the incorporation of monomers such as triethylene glycol dimethacrylate (TEGDMA), urethane dimethacrylate (UDMA), and bisphenol A ethoxylate dimethacrylate (Bis-EMA) enhanced the flowability of the resin, while the addition of micro-, nano-, and hybrid-sized filler particles significantly improved its mechanical and optical properties [3].

Currently, nano-filled restorative materials, known as ‘universal RC’, are widely used. Thanks to their optical compatibility with the surrounding tooth structure, these materials can reduce the visibility of restoration margins (blending effect) and reflect the color of the surrounding tissue (chameleon effect). It has been reported that globular-shaped nanoparticles scatter light in a controlled manner, thereby enhancing optical adaptation [4]. Szalóki et al. [5] showed that gold-patchy silica nanoparticles improved the uniformity of light scattering and optical performance in dental resins, while Fathy et al. [6] reported that graphene oxide nanoparticles modified the optical behavior of nano-filled RCs and influenced color stability. These findings suggest that the initial optical advantages of RCs are related to structural features at the nanoparticle level, but exposure to acidic and oxidizing conditions in the clinical environment may compromise this stability. Nevertheless, the number of studies investigating the effects of various solutions on the physical properties of universal RC, such as roughness and color, remains limited [7].

Other resin-based materials commonly applied include resin-nanoceramic (RNC) and polymer-infiltrated ceramic network (PICN) materials. These are fabricated using computer-aided design/computer-aided manufacturing (CAD/CAM) technology. PICN consists of an inorganic phase (86 wt%) and an organic phase (14 wt%) and is produced by incorporating monomers such as UDMA, Bis-GMA, and TEGDMA into a ceramic network composed of feldspar, zirconia, and sodium aluminosilicate [8]. By contrast, RNCs consist of 71 wt% ceramic fillers and contain nanosilica and zirconia particles that are uniformly dispersed throughout the resin matrix [9].

Regardless of the material used, all dental restoration surfaces should be rendered as smooth as possible before cementation. This is because dental plaque accumulates readily on rough surfaces, thereby increasing the risk of gingival inflammation and secondary caries. Additionally, pigments in liquids can easily adhere to rough surfaces, causing color changes in restorations over time [10]. These changes can be categorized into two main groups: intrinsic and extrinsic. Internal factors include material structure, degree of polymerization, and water absorption, while external factors include contact with beverages, bleaching gel, or mouthwash. It has been well established that organic acids, such as citric and acetic acids, found in liquids, as well as coloring pigments, alcohols, and other chemical compounds, increase color changes [11]. Apple cider vinegar (ACV), containing 4–8% acetic acid, has been reported to possess antimicrobial effects and to dissolve dental plaque, and has been proposed as a potential mouthwash ingredient [12,13]. However, previous studies on ACV have mainly focused on natural tooth structures. One study reported that when ACV was used as an irrigant in root canal treatment, it softened root dentin and increased surface roughness [14]; another examined its effects on enamel demineralization [15]. These investigations were limited to short-term exposures and did not provide sufficient evidence regarding the long-term effects of ACV on modern CAD/CAM restorative materials.

Chlorhexidine-containing mouthwashes are among the most commonly used solutions for preventing dental plaque formation. However, chlorhexidine may degrade into para-chloroaniline, which has been associated with discoloration of dental restorations [16]. These mouthwashes are available in both alcohol-containing and alcohol-free formulations. In the alcohol-containing forms, the softening effect of alcohol on the surface has been reported to further increase roughness and discoloration [17,18]. Moreover, such formulations may cause mucosal irritation and a burning sensation in the mouth, which poses clinical limitations particularly for children and individuals who abstain from alcohol [19]. These limitations have led to the development of alternative formulations, and herbal mouthwashes have been introduced to both address the commercial demand for a broader patient population and respond to the increasing interest in natural ingredients [20,21]. These products are also available in both alcohol-containing and alcohol-free forms. Alcohol-free, water- or glycerin-based herbal formulations are particularly advantageous for children, pregnant women, or individuals with alcohol sensitivity [22]. Nevertheless, due to the presence of natural pigments in their ingredients, herbal mouthwashes have also been reported to have the potential to induce discoloration of restorations [21,23]. Current evidence regarding these products remains limited, and further studies are needed to clarify their effects on the color and physical properties of dental restorations.

Tooth bleaching systems can affect the physical properties of dental restorations due to the oxidative effects of carbamide peroxide (CP) and hydrogen peroxide (HP) [4]. One study [24] reported that the uncontrolled release of free radicals increased surface roughness and compromised the color stability of dental tissues. Another study [25] indicated that conventional HP-based systems may cause soft tissue irritation, highlighting the need for safer carrier systems. Moreover, these systems have been shown to increase surface roughness and induce discoloration not only in teeth but also in CAD/CAM restorative materials [26,27]. Such limitations have prompted the development of new formulations with controlled release properties. One such system, Hydrogen Peroxide Superior (HPS), contains stabilized and encapsulated 25% HP, poloxamer, and other auxiliary components. Thanks to its carrier agents, HPS enables controlled radical release and is applied clinically using a pen-type applicator [28]. However, current research on this new formulation remains limited [29,30].

In recent years, fiber-reinforced RCs (FRCs) have become an important focus of research due to their potential to enhance the mechanical performance of restorations. Fiber strips embedded within the resin matrix have been reported to absorb occlusal forces and thereby improve fracture resistance [31,32]. The type of fiber (glass or polyethylene), its position, density, and adhesion to the resin matrix are considered critical determinants of these mechanical effects [33]. However, investigations addressing the influence of fiber strips on the color stability and surface properties of RCs remain scarce. Previous research [34,35] evaluated the effect of aging on the color stability of FRCs but did not consider the role of fiber strip location. In another study, the surface wear and mechanical properties of FRCs were examined in an orthodontic context; however, fiber position was not included as a variable. Although prior studies have investigated the influence of fiber type or reinforcement on the color stability of RCs, none have addressed the effect of fiber strip location (superficial vs. mid-layer) on optical parameters. Therefore, the present study is the first to evaluate fiber position as an independent variable in relation to color compatibility and whiteness index.

The color stability of dental materials is evaluated using various parameters. Currently, one of the methods that most accurately represents the sensitivity of the human eye to color differences is the CIEDE2000 color change formula (∆E_00_) [36,37]. Additionally, the Whiteness Index change (ΔWI_D_) has gained importance as a complementary parameter for objectively evaluating results obtained with bleaching [38].

In recent years, engineering sciences have proposed models to quantitatively link preparation technology with design development [39]. A similar perspective is adopted in this study. Specifically, preparation techniques such as CAD/CAM-based fabrication, fiber strip positioning, and surface conditioning are decisive in determining the functional outcomes of restorative materials, including surface roughness, color stability, and Whiteness Index. Within this framework, the present study design aims to establish a methodological connection between preparation technology and restoration performance.

For these reasons, this study aims to evaluate the effects of ACV, various mouthwashes, and a bleaching agent on the surface roughness and color stability of RC, PICN, and RNC materials reinforced and non-reinforced with fiber strip. The null hypothesis of the study is that ACV, mouthwashes, and the bleaching agent would not affect the surface roughness, ∆E_00_, and ∆WI_D_ values of the tested materials.

## 2. Materials and Methods

### 2.1. Design of the Study

This study was performed entirely in vitro without the involvement of human participants or animal subjects; thus, ethical approval was not required. Five groups were formed based on the materials used. The materials were as follows: RNC (Cerasmart 250; GC Corporation, Tokyo, Japan [CS]), PICN (Vita Enamic; VITA Zahnfabrik, Bad Säckingen, Germany [VE]), and nanohybrid RC (Filtek Universal; 3M ESPE, St. Paul, MN, USA [FU]). The group with a fiber strip in the middle layer of the FU material was named FUM, and the group with a fiber strip in the upper layer was named FUS. The fiber strip used (everStick NET; GC Corp., Tokyo, Japan [ES]) was a glass fiber-based strip material.

Each group was divided into six subgroups based on the immersion solutions. The solutions used were as follows: artificial saliva (Testonic Artificial Saliva; Colin Kimya, Istanbul, Turkey [AS]), ACV (Kühne Apple Vinegar; Carl Kühne KG, Hamburg, Germany), chlorhexidine-containing and alcohol-containing mouthwash (Andorex Mouthwash; Humanis Health, Istanbul, Turkey [CXA]), chlorhexidine-containing and alcohol-free mouthwash (Klorhex Plus Mouthwash; Drogsan GmbH, Ankara, Turkey [CX]), herbal and alcohol-containing mouthwash (One Drop Only; One Drop Only GmbH, Berlin, Germany [HRA]), herbal and alcohol-free mouthwash (Agarta Mouthwash; Agarta Cosmetics, Ankara, Turkey [HR]). The materials used in the study are presented in Table 1, and the study design is illustrated in Figure 1.

The sample size was determined using a power analysis software (GPower v3.1.9.7; Heinrich-Heine-Universität, Düsseldorf, Germany). The effect size was chosen as a medium effect (f = 0.25; corresponding to η^2^ ≈ 0.06) according to statistical conventions [41] and was found to be consistent with previous studies on CAD/CAM restorative materials [42,43]. In this calculation, the clinically relevant thresholds for Ra, ΔE_00_, and ΔWI_D_ considered in this study were also taken into account. Based on these considerations, a total sample size of n = 300 with a power of 0.95 and α = 0.05 was calculated. Two additional specimens were included in each subgroup for scanning electron microscopy (SEM) analysis. One specimen was designated for examination after immersion in the solutions, while the other was allocated for analysis following the bleaching process.

Specimens were randomly distributed into groups for standardization and randomization. Specimens to be selected for SEM were randomly selected from the groups. Specimen preparation, immersion in solutions, bleaching, surface roughness, and color measurements were performed sequentially. To enhance the reliability of the study, procedures were randomly assigned among researchers, and each procedure was carried out by a different researcher.

### 2.2. Preparation of Specimens

The materials used in the study and the test procedures applied are in accordance with the EN ISO 6872:2024 standard reported for dental restorations [44]. CS and VE blocks were cut to a thickness of 2 mm using a water-cooled specimen cutting device (Micracut 201; Metkon Instruments, Bursa, Turkey). Since the width and length measurements of the blocks were standardized by the manufacturer, no further processing was performed on these dimensions. The specimens were polished using a ceramic polishing set (Diacomp Plus HP; EVE Ernst Vetter GmbH, Keltern, Germany).

The FU specimens were prepared by placing the RC material into a metal mold designed for making 2 mm thick specimens measuring 8 mm × 10 mm. For the FUM group, the RC material was applied to the mold at a thickness of approximately 1 mm. Depth standardization was verified using a periodontal probe. The ES material was cut to a length of 8 mm, placed on the RC material, and condensed using a hand instrument. A second RC layer, 1 mm thick, was then placed on top and similarly condensed using the same instrument (Figure 2).

The only difference between the FUS group and the FUM group was that the ES was positioned 1.5 mm above the RC base. The prepared FU, FUM, and FUS specimens were compressed between two glass plates in the mold to obtain their final shape (Figure 3). Polymerization was performed using an LED curing unit (Elipar Deep Cure S10; 3M ESPE, St. Paul, MN, USA) for 20 s. The specimen surfaces were polished using a RC polishing set (Diacomp Plus HP; EVE Ernst Vetter, Keltern, Germany). All specimens were maintained in distilled water at 37 °C for 24 h in an incubator (UM 400; Memmert GmbH, Schwabach, Germany) to ensure standardization.

### 2.3. Immersion in Solutions and Bleaching

Roughness and color measurements were performed at four different time points: baseline (T0), 1 day after immersion in the solutions (T1), 2.5 days after immersion in the solutions (T2), and after the bleaching process (T3). The specimens were maintained at 37 °C in the same incubator throughout the entire experimental process to simulate clinical conditions.

Specimens were removed from distilled water, rinsed, and gently dried with absorbent paper. Then, T0 measurements were performed. The solutions were left to stand in 15 mL glass jars labeled with specific codes. After placing the specimens in their respective solutions, surface measurements were performed in the T1 phase using the same procedure. The specimens were then returned to their containers, and measurements were performed in the T2 phase using the same method.

An accelerated aging model was applied to simulate clinical conditions [29]. Mouthwashes are typically used for about 2 min per day; therefore, 12 h of continuous immersion was considered equivalent to 1 year of clinical use [43]. Based on this calculation, 24 h and 60 h (2.5 days) of immersion were used to represent approximately 2 years (T1) and 5 years (T2) of clinical exposure, respectively. This model was consistent with previous reports that applied similar accelerated immersion protocols [45,46]. To prevent contamination and maintain a constant concentration, all solutions in which the specimens were stored were renewed daily.

The CXA, CX, and HR solutions were used directly, while the HRA solution was prepared by adding one drop to 15 mL of distilled water, as instructed by the manufacturer. To simulate the oral environment as closely as possible, the pH of the AS solution was adjusted to neutrality. Therefore, an AS solution that complied with the ISO 7491:2000 standard was used [40].

A combined glass electrode pH meter (InoLab pH 720; WTW, Weilheim, Germany) was used to measure the pH values of the solutions [47]. The pH of ACV was diluted to 4.1 with distilled water, in accordance with previous studies [48,49]. Before each test day, the device was calibrated using pH 4 and pH 7 buffer solutions (Certipur Buffer Solution; Merck KGaA, Darmstadt, Germany). During measurements, the glass electrode was placed in 15 mL of solution, and the values were recorded. After each measurement, the probe was rinsed with distilled water and dried.

In the T3 stage, 25% HPS gel (Cavex Bite&White In-Office; Cavex Holland BV, Haarlem, The Netherlands) was applied to the specimen surfaces using the provided applicator pen, according to the manufacturer’s recommendations [28], and the specimens were bleached for 45 min. Subsequently, the specimens were rinsed with distilled water, gently dried with absorbent paper, and final measurements were recorded (Figure 3).

### 2.4. Color and Surface Measurements

Surface roughness measurements were performed using a profilometer (Perthometer M2; Mahr GmbH, Göttingen, Germany). The measurement range was set to 5.6 mm, with a cut-off value of 0.25 mm. A scanning probe of the NHT-6 type with a measurement range of 100 µm was used. Measurements were performed at three points near the center of each specimen, and the mean of the recorded values was calculated. The device was calibrated after each set of ten measurements, taking into account the number of subgroups.

Color measurements were performed in a specially prepared color booth to standardize environmental conditions and prevent errors caused by ambient light. The interior surfaces of the booth, designed with dimensions of 30 cm × 130 cm × 70 cm, were covered with neutral gray cardboard [50]. Measurements were performed using a spectrophotometer (Vita Easyshade V; VITA Zahnfabrik, Bad Säckingen, Germany) (Figure 3). The device was calibrated after each set of ten measurements. To ensure accuracy, each specimen was measured three times, and the mean of these values was calculated.

The interpretation of color change was performed for the T0–T1, T0–T2, and T2–T3 intervals. ∆E_00_ was calculated using the following formula [51]:∆E00=∆L′KLSL2+∆C′KCSC2+∆H′KHSH2+RT∆C′KCSC∆H′KHSH12

ΔL′, ΔC′, and ΔH′ represent changes in brightness, saturation, and hue, respectively. These parameters have been corrected using weighting (S_L_, S_C_, S_H_) and parametric (K_L_, K_C_, K_H_) factors. A rotation factor (R_T_) has been included in the formula to overcome deficiencies in the blue-violet spectrum. In this study, the parametric factors (K_L_, K_C_, and K_H_) were set to 1 [52].

The Whiteness Index (WI_D_) was used to determine the direction of color change. The interpretation of changes in the T0–T1, T0–T2, and T2–T3 intervals was performed using the ∆WI_D_ formula. The formulas are as follows [51]:$${WI}_{D}=0.511L-2.324a-1.100b$$$${∆WI}_{D}={WID}_{treatment}-{WID}_{baseline}$$

The specimens reserved for SEM analysis were subjected to the same immersion and bleaching protocol as the specimens in the relevant solution groups. SEM analyses were performed at stages T2 and T3 to evaluate the effect of the solutions and the bleaching process on surface morphology. Specimens associated with solution groups having the highest average surface roughness (Ra) values were selected for SEM analysis [53]. One of the two reserved specimens was randomly selected and examined at stage T2. The other specimen was examined at stage T3. The same specimen could not be used at both time points because surface coating during the first analysis rendered it unsuitable for subsequent measurement.

Analyses were performed using an SEM instrument (ZEISS EVO 40; Carl Zeiss AG, Oberkochen, Germany). All specimen surfaces were coated with gold before imaging. Each specimen was imaged at magnifications of 5000×, 10,000×, and 25,000×. Images were acquired with the center of the specimens as the reference point. The EHT value was set to 20 kV during the imaging process.

### 2.5. Statistical Analysis

Statistical analyses of roughness and color measurements were performed using software (IBM SPSS Statistics v28.0; IBM Corporation, Armonk, NY, USA). The main effects and interactions of the independent variables (material, solution, and time) were assessed using a three-way analysis of variance (ANOVA). The homogeneity of variances was assessed using Levene’s test, and the normality of the data distribution was evaluated using the Shapiro–Wilk test. Comparisons between significant variables were evaluated using the Bonferroni post hoc test. In all statistical analyses, *p* < 0.05 was considered significant.

## 3. Results

### 3.1. Surface Roughness Analysis

The ANOVA results for Ra values are shown in Table 2. The results revealed statistically significant effects of material, solution, time, and solution-time on Ra (*p* < 0.001).

When the material was examined regardless of solution or time, the highest value was found in CS (0.30 µm) and the lowest in VE (0.22 µm). For the solution, the highest value was found in ACV (0.31 µm) and the lowest in AS (0.20 µm). For time, the highest value was found in T3 (0.33 µm) and the lowest in T0 (0.17 µm) (*p* < 0.001) (Table 3).

When the material-solution interaction was evaluated regardless of time, ACV generally caused the greatest change in Ra in all materials, while AS caused the lowest (*p* > 0.05). The highest value was found for CS–ACV (0.36 µm), while the lowest value was for VE–AS (0.18 µm) (*p* > 0.05).

When the material-time interaction was evaluated regardless of solution, T3 generally caused the greatest change in Ra in all materials, while T0 caused the lowest (*p* > 0.05). The highest value was found for CS–T3 (0.39 µm), while the lowest value was for FUM–T0 (0.16 µm) (*p* > 0.05) (Figure 4).

When the solution-time interaction was evaluated regardless of material, T3 generally caused the greatest change in Ra in all solutions, while T0 caused the lowest (*p* < 0.001). The highest value was found for ACV–T3 (0.43 µm), and the lowest was in ACV–T0 (0.16 µm) (*p* < 0.001).

When the material-solution-time interaction was evaluated, Ra generally increased from T0 to T3 for all materials in each solution. The highest value was found in the CS–ACV–T3 solution (0.47 µm), while the lowest value was found in the FU–ACV–T0 solution (0.13 µm) (*p* > 0.05).

### 3.2. ∆E_00_ Analysis

The ANOVA results for ∆E_00_ values are shown in Table 2. The results revealed statistically significant effects of material, solution, time, material-solution, material-time, and solution-time on ∆E_00_ (*p* < 0.001).

When the material was examined regardless of solution or time, the highest value was found in FU (ΔE_00_ = 1.57) and the lowest in VE (ΔE_00_ = 0.81). For the solutions, the highest value was found in ACV (ΔE_00_ = 1.66) and the lowest in HR (ΔE_00_ = 0.57). For time, the highest value was found in T0–T2 (ΔE_00_ = 1.53) and the lowest in T2–T3 (ΔE_00_ = 0.64) (*p* < 0.001) (Table 4).

When the material-solution interaction was evaluated regardless of time, ACV and CXA caused the greatest change in ∆E_00_, while AS and HR caused the lowest (*p* < 0.001). The highest value was in FU–ACV (ΔE_00_ = 1.95), and the lowest was in VE–HR (ΔE_00_ = 0.24) (*p* < 0.001).

When the material-time interaction was evaluated regardless of solution, T0–T2 generally caused the greatest change in ∆E_00_ in all materials, while T2–T3 caused the lowest (*p* < 0.001). The highest value was found in FU–T0–T2 (ΔE_00_ = 2.15), and the lowest in VE–T2–T3 (ΔE_00_ = 0.53) (*p* < 0.001) (Figure 4).

When the solution-time interaction was evaluated regardless of material, T0–T2 generally caused the greatest change in ∆E_00_ in all solutions, while T2–T3 caused the lowest (*p* < 0.001). The highest value was found in the CXA–T0–T2 solution (ΔE_00_ = 2.20), and the lowest was in the HR–T2–T3 solution (ΔE_00_ = 0.34) (*p* < 0.001).

When the material-solution-time interaction was evaluated, ∆E_00_ generally showed the highest values in T0–T2 and the lowest in T2–T3 for all materials in each solution (*p* > 0.05). The highest value was found in the FU–ACV–T0–T2 combination (ΔE_00_ = 2.68), while the lowest value was found in the VE–HR–T0–T1 combination (ΔE_00_ = 0.18) (*p* > 0.05).

### 3.3. ∆WI_D_ Analysis

The ANOVA results for ∆WI_D_ values are shown in Table 2. The results revealed statistically significant effects of material, solution, time, material-solution, material-time, and solution-time on ∆WI_D_ (*p* < 0.001).

When the material was examined regardless of solution or time, the highest change was found in FU (ΔWI_D_ = −0.68) and the lowest in VE (ΔWI_D_ = −0.40). For the solutions, the highest value was found in ACV (ΔWI_D_ = −0.90) and the lowest in HR (ΔWI_D_ = −0.06). For time, the highest value was found in T0–T2 (ΔWI_D_ = −1.20) and the lowest in T2–T3 (ΔWI_D_ = 0.43) (*p* < 0.001) (Table 5).

When the material-solution interaction was evaluated regardless of time, ACV generally caused the greatest change in ∆WI_D_, while HR and HRA generally caused the lowest (*p* < 0.001). The highest value was in FU–ACV (ΔWI_D_ = −1.18), and the lowest in VE–HRA (ΔWI_D_ = −0.04) (*p* < 0.001).

When the material-time interaction was evaluated regardless of solution, T0–T2 caused the greatest change in ∆WI_D_ in all materials, while T2–T3 caused the lowest (*p* < 0.001). The highest change was found in FU–T0–T2 (ΔWI_D_ = −1.46), and the lowest in VE–T2–T3 (ΔWI_D_ = 0.32) (*p* < 0.001).

When the solution-time interaction was evaluated regardless of material, T0–T2 caused the greatest change in ∆WI_D_ in all solutions, while T2–T3 generally caused the lowest (*p* < 0.001). The highest change was found in CXA–T0–T2 (ΔWI_D_ = −1.87), and the lowest was in HR–T0–T1 (ΔWI_D_ = −0.19) (*p* < 0.001) (Figure 4).

When the material-solution-time interaction was evaluated, ∆WI_D_ generally showed the highest changes in T0–T2 for all materials and the lowest in T2–T3 (*p* < 0.001). The highest change was found in the FU–ACV–T0–T2 combination (ΔWI_D_ = −2.38), while the lowest change was found in the VE–AS–T0–T1 combination (ΔWI_D_ = 0.02) (*p* < 0.001).

### 3.4. SEM Analysis

Since the highest Ra value for FU was found in the ACV group at T2, the FU–ACV specimen was analyzed. At 5000× magnification at T2, deepened polishing lines, fine pitting, filler protrusions, matrix irregularities, and partially exposed filler particles were observed on the surface (Figure 5a). At 10,000× magnification, debonding at the matrix–filler interface and localized voids corresponding to detached filler particles were evident (Figure 5b). At 25,000× magnification, nanoscale interfacial debonding and fine microcracks propagating throughout the resin matrix were detected. At 25,000× magnification, increased matrix roughness and the presence of localized voids also became more pronounced (Figure 5c).

At T3, the surface appeared generally homogeneous at 5000× magnification. However, numerous small pits and fine linear traces were present, indicating localized matrix dissolution and superficial wear (Figure 6a). At 10,000× magnification, pronounced micropitting across the surface and localized debonding at the filler–matrix interface were evident. These findings suggested partial dislodgement of filler particles and the onset of early microstructural degradation (Figure 6b). At 25,000× magnification, the surface exhibited nanopitting, shallow groove-like microcracks, and a granular roughness pattern distributed across the matrix. These nanoscale irregularities reflected localized matrix erosion and interfacial stresses at the filler–matrix junction (Figure 6c).

Since the highest Ra value for FUS was found in the ACV group at T2, the FUS–ACV specimen was analyzed. At T2, shallow to moderate pitting and localized matrix degradation were observed at 5000× magnification. These findings were accompanied by parallel polishing lines and distinct grooves. In some areas, fibers appeared closer to the surface, and partially exposed filler clusters were evident along the grooves (Figure 5d). At 10,000× magnification, localized voids and interfacial gaps between the resin matrix and filler particles were observed. Fine microcracks originating from these defects were also detected, and some filler particles appeared partially detached from the matrix (Figure 5e). At 25,000× magnification, numerous cracks, granular roughness, and signs of matrix erosion were observed across the surface. In certain regions, the matrix appeared porous and fragmented, which indicated progressive surface degradation (Figure 5f).

At T3, pronounced cracks, fractured areas, and interfacial separations were observed at 5000× magnification. Sharp edges along the fracture lines indicated matrix fragmentation and localized detachment of the resin from the surface (Figure 6d). At 10,000× magnification, fragmented matrix structures with irregular voids and localized detachment areas were evident. Numerous fractured areas demonstrated a substantial weakening of the resin–filler interface (Figure 6e). At 25,000× magnification, pronounced pitting, localized separations, and sharp-edged surface irregularities were observed, indicating advanced disruption of the resin matrix (Figure 6f).

As the highest Ra value for FUM was found in the CXA group at T2, the FUM–CXA specimen was analyzed. At T2, interfacial debonding between the resin matrix and filler particles, together with distinct linear traces across the surface, were observed at 5000× magnification. Localized roughened areas and shallow pits were also evident, indicating the initial stages of surface degradation (Figure 5g). At 10,000× magnification, fine surface irregularities and interfacial debonding zones were noted. Small voids and shallow microcracks were also detected, suggesting progressive weakening of the matrix–filler interface (Figure 5h). At 25,000× magnification, granular roughness, shallow pits, and nanoscale surface irregularities were identified. Localized porosities and early microvoids were also observed, indicating progressive weakening of the resin matrix (Figure 5i).

At T3, scattered small pits and faint linear traces reflecting limited surface disruption and early matrix alteration were observed at 5000× magnification (Figure 6g). At 10,000× magnification, fine grooves, distinct interfacial separations, and localized wear areas were evident, indicating progressive but moderate surface degradation (Figure 6h). At 25,000× magnification, nanopitting, interfacial separations, and a granular roughness pattern were observed, reflecting nanoscale degradation of the resin matrix (Figure 6i).

Since the highest Ra value for CS was found in the ACV group at T2, the CS–ACV specimen was analyzed. At T2, the surface appeared divided by distinct linear traces at 5000× magnification. Along these grooves, localized pitting, matrix wear, and early signs of filler protrusion were evident (Figure 5j). At 10,000× magnification, widespread microdebonding between the resin matrix and filler particles was observed. Sharp-edged voids and irregular boundaries were detected, indicating progressive filler loss and localized matrix collapse (Figure 5k). At 25,000× magnification, numerous nanoscale pits, signs of matrix erosion, and granular surface roughness were observed. The surface appeared porous and fragmented, with irregular contours and scattered microvoids indicating advanced degradation of the resin–filler interface (Figure 5l).

At T3, a roughened surface topography with distinct linear traces was observed at 5000× magnification, indicating the onset of peroxide-induced surface wear (Figure 6j). At 10,000× magnification, widespread micropitting and early signs of matrix–filler debonding were evident, suggesting peroxide-related weakening of the composite interface (Figure 6k). At 25,000× magnification, granular roughness and pronounced erosion of the matrix were observed, indicating advanced peroxide-induced surface degradation (Figure 6l).

Since the highest Ra value for VE was found in the ACV group at T2, the VE–ACV specimen was analyzed. At T2, the 5000× SEM image revealed irregular ceramic network regions with sharp edges. Distinct cracks traversed the surface, and in some areas, the polymer phase was detached from the ceramic framework. These features indicated structural incompatibility between the polymer and ceramic phases under acidic exposure (Figure 5m). At 10,000× magnification, distinct voids caused by polymer phase loss, as well as prominent gaps between plate-like ceramic structures, were evident. The ceramic phase appeared fractured and irregular, and in several areas, separation of the resin phase was detected (Figure 5n). At 25,000× magnification, multiple microcracks crossing the ceramic–polymer interface, together with deep pits and fragmented edges, were observed. The surface appeared severely disrupted, with sharp contours and material loss, suggesting advanced acid-induced degradation (Figure 5o).

At T3, the 5000× SEM image revealed a ceramic network structure with distinct deep voids, reflecting surface degradation in the polymer-infiltrated regions (Figure 6m). At 10,000× magnification, voids within the matrix were evident, along with cracks and sharp-edged ceramic fragments resulting from polymer loss (Figure 6n). At 25,000× magnification, widespread microcracks, sharp-edged pits, and fractured margins were observed. The ceramic network appeared highly irregular due to prominent cracks extending between the ceramic and polymer phases, accompanied by fragmented areas and interfacial gaps (Figure 6o).

## 4. Discussion

Based on the statistical analysis results obtained in this study, the null hypothesis, stating that ACV, mouthwashes, and bleaching agents would not affect the surface roughness, ∆E_00_, and ∆WI_D_ values of the tested materials, was rejected. Previous studies have reported the Ra threshold for bacterial adhesion as 0.2 µm and the perceptibility threshold for the tongue as 0.5 µm [54,55]. In this study, Ra values were below 0.2 µm at T0, increased at later time points, but all groups remained below 0.5 µm.

In this study, surface roughness values increased considerably in acidic or alcohol-containing solutions such as ACV, CXA, and HRA. At T3, Ra values reached 0.43 µm in FU–ACV, 0.45 µm in FUS–ACV, 0.47 µm in CS–ACV, 0.44 µm in CS–CXA, and 0.42 µm in CS–HRA, whereas the baseline values remained below the 0.2 µm threshold. In the CX group, Ra increased to 0.34 µm at T3, exceeding the 0.2 µm threshold but remaining lower than ACV and CXA. In contrast, neutral solutions such as AS and HR maintained Ra values below 0.29 µm throughout the study. Of note, although several groups exceeded the 0.2 µm bacterial adhesion threshold, none reached the 0.5 µm acceptability threshold. These findings indicate that exposure to acidic and alcohol-containing solutions in combination with oxidizing agents can elevate surface roughness to clinically acceptable levels yet may still have the potential to induce biological effects. From a clinical standpoint, these findings indicate that even if color stability remains within acceptable limits, increased surface roughness may pose risks for biofilm retention, secondary caries, and periodontal inflammation. Such risks can be mitigated through regular professional maintenance and repolishing procedures. Moreover, careful material selection and patient instruction in cases with frequent exposure to acidic or alcohol-containing solutions are crucial for the long-term success of restorations.

Roughness measurements are commonly performed using contact-type profilometers; however, the path followed by the device during measurement may influence the Ra value [56]. In this study, three measurements were performed on each specimen, and the mean values were calculated to improve measurement reliability. It has been reported that differences in polishing techniques may affect the Ra value [57]. Therefore, to standardize the process, the specimens were polished using a method consistent with the manufacturer’s recommendations.

Vinegar contains 4–8% acetic acid, which can increase roughness by softening the matrix, displacing filler particles, and promoting water absorption [58,59]. However, vinegar has been reported to indirectly reduce the risk of gingival infection by inhibiting dental plaque formation [12,47]. Research has shown that the effects of vinegar on tooth enamel and dental restorations can be studied in vitro, but these effects cannot be directly applied to in vivo results because of various factors, including saliva, pellicle, feeding time, buffering, and brushing [60]. Therefore, ACV was tested in vitro in this study.

For ACV to be used as a mouthwash, its pH must be above a critical threshold. According to the ISO 16408:2015 standard, the pH range for mouthwashes is 3.0–10.5 [48,61]. In studies on dental erosion, the pH is commonly kept within the range of 4.0–4.5 [49,62]. Furthermore, the pH range of most commercial mouthwashes has been reported to be 4.1–7.9 [48]. Therefore, in this study, the pH was adjusted to 4.1 to simulate the acidic effect of ACV.

In this study, AS with a pH of 6.8 was used, in accordance with ISO 7491:2000 [40]. This value was chosen to mimic the average pH of unstimulated saliva. When the solutions were examined, regardless of the material, the order of Ra in T2 was ACV > CXA > CX > HRA > HR > AS. Consistent with a previous study [63], roughness increased by only about 0.02 µm in the AS group. This minimal change is consistent with the acid-buffering effect of saliva.

The recommended pH threshold for preventing tooth or material erosion is 5.5 [64]. Of the mouthwashes used in this study, only CXA was below this threshold, with a pH level of 5.2. Consistent with a previous study [21], this study found that alcohol-containing mouthwashes increased Ra more than alcohol-free mouthwashes. Furthermore, it was found that herbal mouthwashes increased Ra by less than mouthwashes with chlorhexidine. Although chlorhexidine is considered the gold standard for dental plaque control, prolonged use may cause deterioration and discoloration of restoration surfaces. Furthermore, alcohol may degrade polymer chains, resulting in surface wear [43].

In this study, consistent with Aksoy Vaizoğlu [65], Ra values increased in specimens exposed to mouthwashes, while the increase was negligible in the HRA and HR groups, possibly due to the limited acidity and solubility of their herbal components. The results also suggest that mouthwashes may potentiate the roughness-increasing effect of bleaching, likely because the weakened matrix is more susceptible to degradation during bleaching. Furthermore, bleaching agents with viscous carriers may remain on the surface for longer, thereby enhancing penetration and exacerbating deterioration [65].

It is well established that HP agents can influence the surface properties of dental restorative materials [4,58]. In this study, Ra values after HPS application remained minimal and below the clinical threshold. This finding indicates that HPS preserved surface integrity due to its controlled radical release. The fundamental difference between acidic exposure and office-type bleaching lies in the duration and chemical activity of contact. Low-pH solutions such as ACV, under continuous immersion, cause swelling of hydrophilic monomers in the resin matrix and debonding of filler–matrix interfaces, leading to a pronounced increase in roughness [66]. In contrast, HPS provides a short-term and controlled application that limits radical activity and results in only minimal surface alterations. Consequently, ACV produced more evident surface degradation compared with HPS, whereas bleaching induced noticeable changes in color and whiteness without substantially compromising surface integrity.

The thermogelation capacity of the HPS carrier system is primarily attributed to its Poloxamer 407 component, which enhances stability during application and protects the material surface from excessive radical effects. From a clinical perspective, this is important for maintaining the color stability of restorations and their resistance to plaque accumulation. Moreover, the unique application method and carrier agent composition of HPS suggest that it may represent a safer and more material-friendly alternative compared with conventional systems [28,29].

∆E_00_ was evaluated according to the clinical perceptibility (PT = 0.8) and acceptability (AT = 1.8) thresholds [67]. A previous study reported that mouthwashes did not exceed the AT threshold in dental materials [68], and these results are consistent with those of the present study. The AT threshold was exceeded in the short and medium term for the following combinations: AS, ACV, CXA, CX, and HRA for FU; ACV and CXA for FU, FUS, and FUM; and CXA for VE.

Consistent with a previous study [64], color change was found above the AT threshold for all materials in contact with CXA at T0–T2. The color change caused by CXA may have been due to the binding of para-chloroaniline, one of the degradation products of chlorhexidine, to the material surface, and the alcohol content facilitating pigment penetration by increasing polymer matrix solubility.

Kader et al. [45] evaluated VE, CS, and RC materials in alcoholic and alcohol-free solutions. Similarly to this study, the researchers reported significant time-dependent color changes and stated that high resin content, in particular, increases the susceptibility to color change. Currently available universal RCs provide optical compatibility with surrounding dental tissues by optimizing light scattering and transmission thanks to their nanofillers [4,7], and can be used in both anterior and posterior regions. The significant color change observed in the nanohybrid RC tested in our study is consistent with previous studies [3,69], which found that TEGDMA content is associated with increased water sorption and susceptibility to discoloration.

Although CS has a high content of inorganic filler, the matrix phase is sensitive to acids, alcohols, and oxidizing agents. This can result in dissolution-induced micropits. VE, on the other hand, is obtained by infiltrating the feldspathic ceramic phase with a low-viscosity polymer. This renders it more resistant than CS, as it has a strong ceramic network and a low polymer content. However, partial dissolution of the polymer phase may cause microporosity in the ceramic matrix. This effect is less pronounced than in materials with a high polymer content, such as FU [70].

The higher color change in CS compared to VE may be attributed to mismatch between the polymer matrix and the refractive index. This is caused by the nanoceramic filler structure of CS, consisting of silica and barium particles, which results in greater light scattering. In FU, the presence of zirconia/silica particles (0.6–1.4 µm) and hydrophilic monomers such as Bis-GMA/TEGDMA within the matrix may cause higher ∆E_00_ values by increasing water absorption. Increasing the TEGDMA ratio from 0% to 1% in Bis-GMA-based RCs has been reported to increase water absorption from 3% to 6%. Water absorption is relatively lower in UDMA-based RC materials [71]. For these reasons, materials containing high levels of hydrophilic monomers, large fillers, and exhibiting poor refractive index compatibility can be expected to exhibit discoloration that exceeds the clinically acceptable limit in low-pH or alcohol-containing solutions.

The artificial or natural colorants contained in mouthwashes may cause color changes in dental restorations. Synthetic colorants, such as Patent Blue V and Tartrazine, which are found in CXA and CX, may have been adsorbed onto the material surface, leading to discoloration. The conversion of chlorophyll from green tea in HRA to pheophytin under acidic or oxidative conditions may also contribute to color change. Similarly, natural pigments from herbal ingredients such as mint, clove, sage, thyme, and tea tree in HR may cause color changes to restorative materials. Furthermore, the effect of natural pigments from apples in ACV should not be ignored [64]. Although no colorant was present in the AS group, the AT value was exceeded in the FU–T0–T2 combination. This may be related to prolonged water sorption and the plasticization of the polymer matrix, as reported in a previous study [68].

Savic-Stankovic et al. [4] reported that bleaching in RCs reduced the roughness to a level that could be eliminated through polishing, resulting in color changes that were clinically perceptible. Similar results were obtained in our study, with specimens immersed in CXA and ACV experiencing greater effects from bleaching than the others. This may be attributed to degradation of the resin matrix phase induced by prior exposure to acidic, alcoholic, or pigmented solutions. Furthermore, the pronounced whitening after bleaching in darker specimens compared to initial conditions may be explained by the oxidative elimination of pigments. However, it should be noted that this effect may vary depending on the material structure [72].

In this study, ∆WI_D_ was evaluated according to clinical perceptibility (WPT) and clinical acceptability (WAT) thresholds of 0.72 and 2.60, respectively [64]. ∆WI_D_ results were consistent with ∆E_00_ results. The WAT threshold was not exceeded in any combination. However, at T0–T2, the WPT threshold was exceeded in the AS, ACV, CXA, CS, and HRA groups for FU; in the ACV, CXA, and CX groups for FUS; in the AS, ACV, CXA, and CX groups for FUM; in all solution groups for CS; and in the ACV, CXA, and CX groups for VE.

In a study examining the effects of home bleaching, Erturk-Avunduk et al. [72] reported ∆WI_D_ values exceeding the WAT threshold, which is different from the results of this study. This suggests that home bleaching may be more effective than office bleaching at removing extrinsic stains. Previous studies [24,26] noted that the effectiveness of bleaching depends on application time and product content. The controlled-release technology of HPS may have prevented the WAT threshold from being exceeded in our study.

In recent years, the reinforcement of RCs with fiber strips has been widely investigated to improve their mechanical performance. Polyethylene (PF) or glass fiber (GF)-based strips absorb local stresses by regulating the force distribution within the resin matrix and increasing fracture resistance [31,32]. The GF strip used in this study exhibited differences depending on its position within the RC. The pre-impregnated structure of ES provides better bonding, a lower void ratio, and limited water sorption compared to PFs. In contrast, in non-pre-impregnated fibers, resin penetration is insufficient. Due to microvoids and high water sorption, discoloration, decreased whiteness, and deterioration of surface integrity may occur over time [73,74]. Moreover, in PFs, where the difference in refractive indices between the fiber and the matrix is pronounced, poor optical conformity may lead to matte areas, whereas in GFs, the refractive index is closer to that of the matrix, leading to better optical compatibility [75].

In this study, the increase in roughness observed in the FUS group may be related to the surface position of the fiber strip, which, despite its pre-impregnated GF structure, was directly exposed to chemical and mechanical effects [74]. In contrast, the fiber strip in the FUM group was embedded within the RC mass; therefore, the increase in Ra was not attributable to fibers protruding onto the surface. When the values of FU, FUM, and FUS were examined at T0–T1 and T0–T2, the order was FU > FUM > FUS. This finding suggests that fibers placed in the mid-layer may cause greater scattering of light within the material, thereby increasing color change, whereas superficial fibers may partially reflect and diffuse light, providing an optical masking effect. The close refractive indices between fiber and matrix may contribute to this phenomenon [76]. Nakamura et al. [77] reported that the similar refractive indices of GFs and the resin matrix facilitated light transmission in GF–reinforced RCs, while Pasmadjian et al. [78] emphasized that GFs enhance light transmission, supporting natural appearance and optical homogeneity.

The primary reason why fiber strip position did not result in a marked difference in surface roughness is that Ra values were mainly determined by matrix–filler interactions occurring under chemical exposure. Even when positioned superficially, the pre-impregnated fibers were integrated within the resin matrix and contributed minimally to the surface topography captured by the profilometer. In contrast, optical parameters are more sensitive to light scattering and reflections within the material, making fiber position a stronger determinant for these outcomes.

One study [34] reported that the color of FRCs gradually changed over time due to microcracks caused by water sorption. PFs were found to have a higher potential for discoloration than GFs, and the superiority of GFs was attributed to their hydrophobic inorganic structures. In addition, a six-year clinical follow-up revealed that the long-term performance may be affected by the quality of the fiber–matrix bond and the geometry of the fibers [70].

The SEM analyses obtained in this study demonstrated interfacial debonding, pitting, and nanocracks at the matrix–particle interface in FU specimens immersed in ACV, consistent with previous findings [69]. In addition, surface irregularities and filler exposure observed in this group were associated with the elevated Ra values, and the increase in ΔE_00_ could be attributed to enhanced light scattering. Softening of the resin matrix in an acidic environment weakens the filler–matrix bond, leading to multiscale structural degradation. Similar surface deformations were also observed in FUM specimens stored in CXA, where fine irregularities and microcracks reflected the accelerated degradation caused by the combined effect of alcohol and chlorhexidine.

Findings from the FUS group indicated that fibers placed superficially are more susceptible to degradation under acidic conditions. However, the increase in ΔE_00_ values in the FUS group was more limited compared with FU, suggesting that superficial fiber placement may enhance light reflection and produce a lighter perceived shade, thereby partially masking the extent of color change. On the other hand, at T3, wide cracks, fragmented matrix structures, and sharp-edged fractured regions observed in FUS specimens exposed to HPS demonstrated the higher susceptibility of superficial fibers to peroxide-induced oxidative stress. In contrast, in the FUM group, where fibers were placed in the mid-layer, only minor pits and superficial irregularities were detected, indicating that this positioning provides a buffering effect that partially preserves surface integrity.

In the CS group, particle shrinkage, granular irregularities, and widespread micro-debonding were observed following ACV exposure. These morphological features explain the increased Ra and ΔE_00_ values and indicate that acetic acid accelerates filler–matrix separation. In CS specimens exposed to HPS, granular roughness and micropitting were evident; although the increase in surface roughness did not exceed the clinical threshold, these findings suggest that long-term stability may still be compromised.

In the VE group, ACV exposure resulted in sharp-edged irregularities in the ceramic network, polymer phase loss, and interfacial voids, highlighting the structural incompatibility between the ceramic and polymer phases under acidic conditions. In specimens exposed to HPS, cracks and sharp-edged ceramic fragments associated with polymer loss were observed, indicating that dissolution of the polymer phase weakens the ceramic framework.

Previous SEM studies have reported that peroxide penetration causes separation at the matrix–particle interface, thereby creating significant heterogeneity in surface topography [29,79]. It has also been suggested that high-concentration HP applications may induce microscopic changes without increasing macroscopic roughness. Moreover, acid exposure has been reported to exacerbate matrix degradation by attacking the fibers [80]. One study [81] reported the exposure of numerous filamentous GFs longer than 50 µm and approximately 20 µm in diameter, noting an inhomogeneous distribution. The statistically higher Ra values observed in fiber strip–reinforced groups compared with non-reinforced groups in the present study support this mechanism. The same study [81] also indicated that structural deterioration and discoloration in fiber strip–reinforced RCs usually occur within 3–6 years, emphasizing the need for long-term evaluations of restorative materials. The SEM findings of the present investigation suggest that degradation mechanisms occurring in vivo over long periods can be simulated in vitro within a shorter timeframe, yielding results consistent with clinical observations.

This study has some limitations. Although the study was conducted under in vitro conditions, which enabled the variables to be controlled, it did not simulate the multiple factors encountered in a clinical setting, such as thermal changes, mechanical forces, salivary flow, and biological interactions. Furthermore, the evaluation was limited to specific material groups and solution types; different manufacturer formulations, alternative solutions, and long-term environmental factors may affect these results differently. Therefore, future studies incorporating in vivo conditions would provide a more comprehensive understanding of the effects on color stability, surface roughness, and structural integrity of restorative materials.

## 5. Conclusions

The results obtained within the limitations of this study are as follows:

Surface roughness: Initial Ra values for all materials were below the bacterial adhesion threshold of 0.2 µm. Ra values increased over time, with the highest increases in ACV and CXA and the lowest in AS and HR groups.

Color change: The highest ∆E_00_ values were observed in FU and FUM with ACV and CXA, whereas VE showed the lowest values. In most combinations, ∆E_00_ values remained below the clinical acceptability threshold.

Whiteness index change: A similar pattern was observed. The highest values were found in FU, FUM, and CS, with ACV and CXA, while VE showed the lowest values. All combinations remained below the clinical acceptability threshold.

Fiber strip: The location of the ES strip within the resin composite did not affect surface roughness but influenced color and whiteness changes. Short- and medium-term changes were higher in the FUM group compared with the FUS group.

Bleaching: The increase in Ra following HPS application was minimal and remained below the clinical perceptibility threshold. Changes in color and the whiteness index also remained below the clinical acceptability threshold.

## Figures and Tables

**Figure 1 polymers-17-02552-f001:**
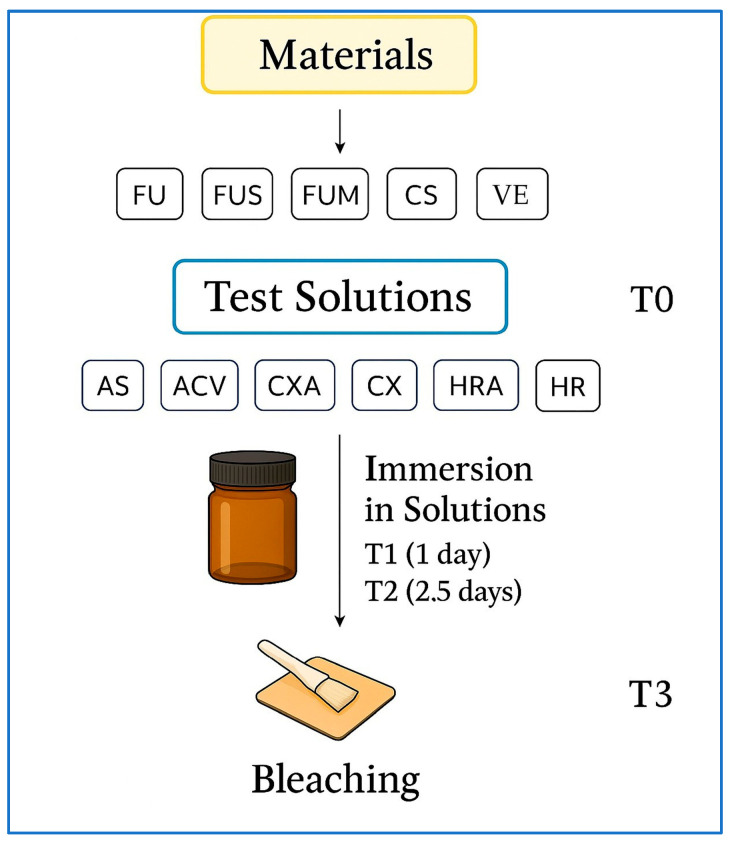
Study design. FU: Filtek Universal; FUS: Filtek Universal with superficial fiber strip; FUM: Filtek Universal with mid-layer fiber strip; CS: Cerasmart 270; VE: Vita Enamic; ES: everStick NET; AS: Artificial saliva; ACV: Apple cider vinegar; CXA: Chlorhexidine- and alcohol-containing mouthwash; CX: Chlorhexidine- and alcohol-free mouthwash; HRA: Herbal and alcohol-containing mouthwash; HR: Herbal and alcohol-free mouthwash; T0: Baseline; T3: After bleaching. Abbreviations used throughout text and figures.

**Figure 2 polymers-17-02552-f002:**
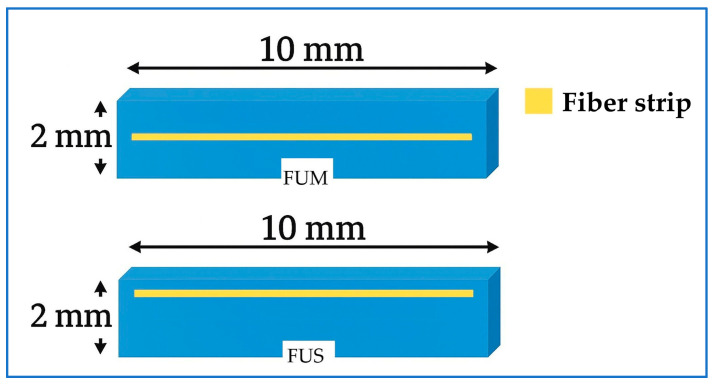
Schematic illustration of fiber strip positions. FUM: nanohybrid RC with mid-layer fiber strip; FUS: nanohybrid RC with superficial fiber strip. Abbreviations used throughout text and figures.

**Figure 3 polymers-17-02552-f003:**
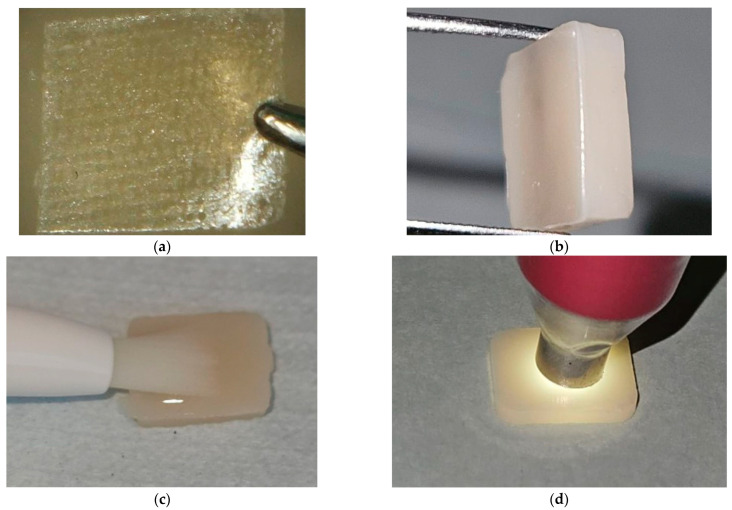
Specimen preparation stages. (**a**) Preparation of the fiber strip for a FUS specimen; (**b**) A prepared FUS specimen; (**c**) Application of the bleaching agent on a nanohybrid RC specimen; (**d**) Color measurement of a CS specimen. FUS: Filtek Universal with superficial fiber strip; RC: Resin composite; CS: Cerasmart 270. Abbreviations used throughout text and figures.

**Figure 4 polymers-17-02552-f004:**
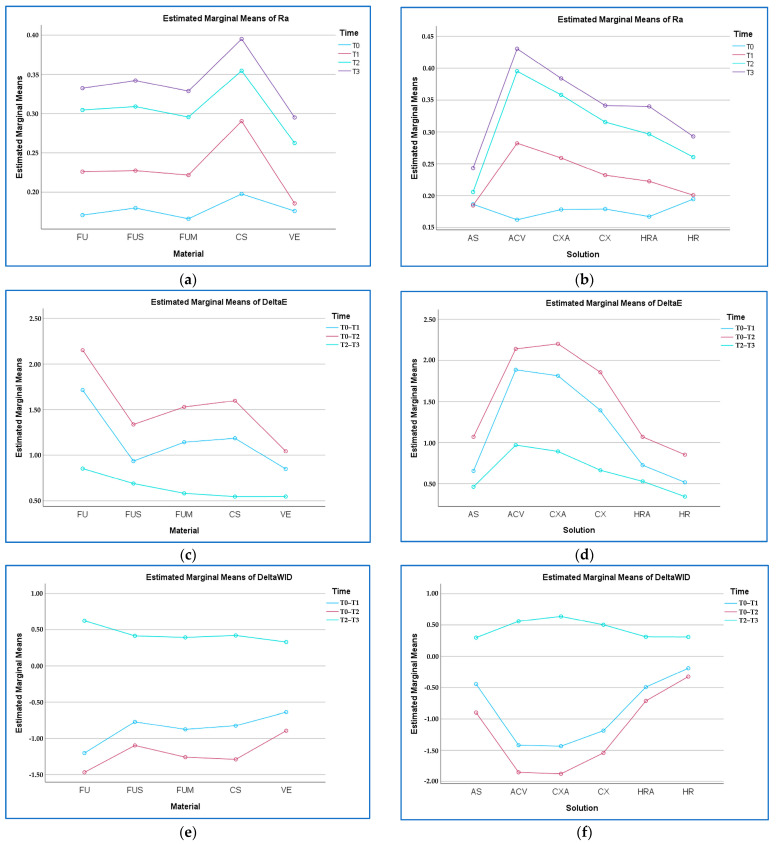
Material–time and solution–time graphs. (**a**) Ra (µm) by material–time; (**b**) Ra by solution–time; (**c**) ΔE_00_ by material–time; (**d**) ΔE_00_ by solution–time; (**e**) ΔWI_D_ by material–time; (**f**) ΔWI_D_ by solution–time. FU: Filtek Universal; FUS: Filtek Universal with superficial fiber strip; FUM: Filtek Universal with mid-layer fiber strip; CS: Cerasmart 270; VE: Vita Enamic; AS: Artificial saliva; ACV: Apple cider vinegar; CXA: Chlorhexidine- and alcohol-containing mouthwash; CX: Chlorhexidine- and alcohol-free mouthwash; HRA: Herbal and alcohol-containing mouthwash; HR: Herbal and alcohol-free mouthwash; T0: Baseline; T3: After bleaching. Abbreviations used throughout text and figures.

**Figure 5 polymers-17-02552-f005:**
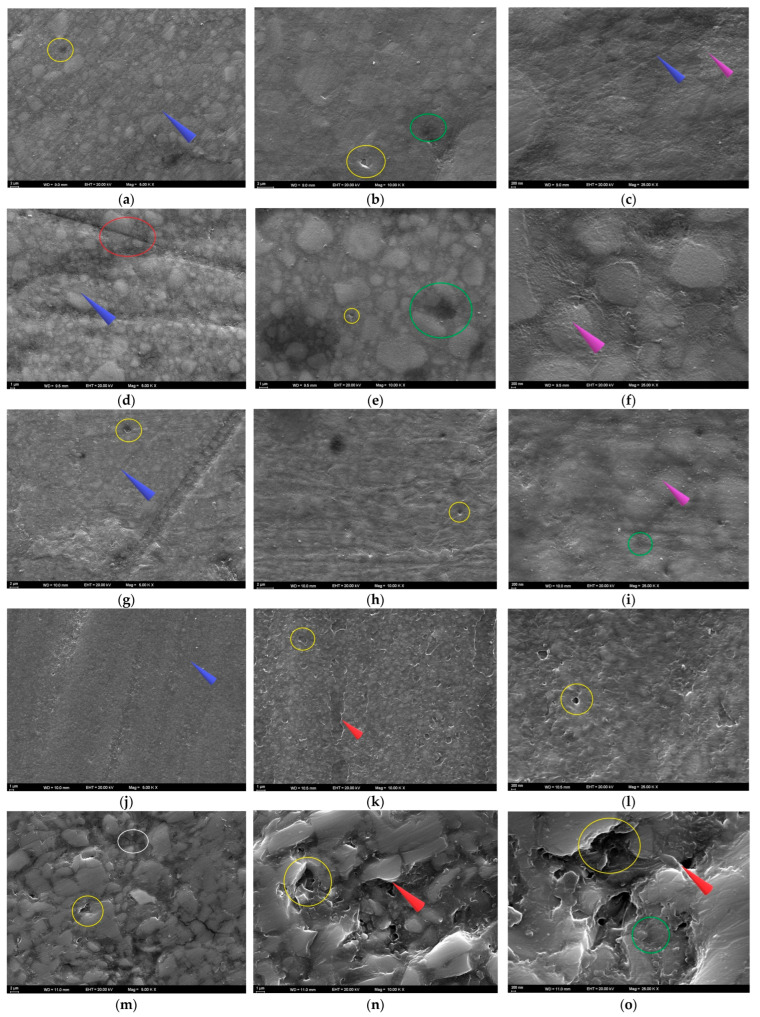
SEM images of the specimens following solution immersion. (**a**–**c**) FU at 5000×, 10,000×, and 25,000×; (**d**–**f**) FUS at 5000×, 10,000×, and 25,000×; (**g**–**i**) FUM at 5000×, 10,000×, and 25,000×; (**j**–**l**) CS at 5000×, 10,000×, and 25,000×; (**m**–**o**) VE at 5000×, 10,000×, and 25,000×. Red circle: groove; blue arrow: linear traces; yellow circle: debonding; green circle: pitting; purple arrow: roughness; white circle: crack; red arrow: sharp edge.

**Figure 6 polymers-17-02552-f006:**
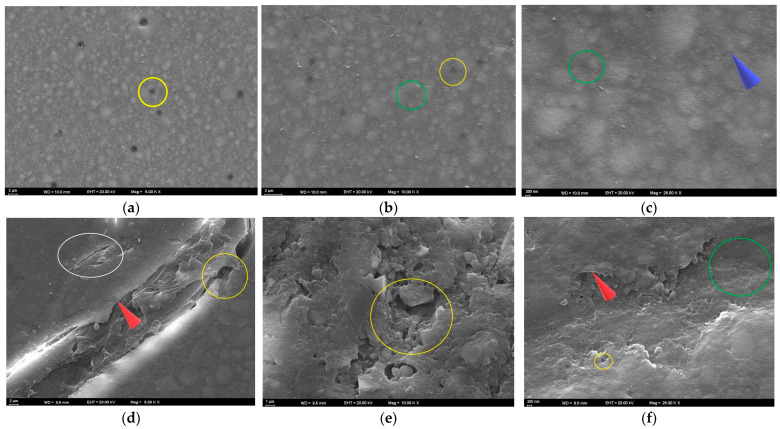
SEM images of the specimens after bleaching. (**a**–**c**) FU at 5000×, 10,000×, and 25,000×; (**d**–**f**) FUS at 5000×, 10,000×, and 25,000×; (**g**–**i**) FUM at 5000×, 10,000×, and 25,000×; (**j**–**l**) CS at 5000×, 10,000×, and 25,000×; (**m**–**o**) VE at 5000×, 10,000×, and 25,000×. Red circle: groove; blue arrow: linear traces; yellow circle: debonding; green circle: pitting; purple arrow: roughness; white circle: crack; red arrow: sharp edge.

**Table 1 polymers-17-02552-t001:** Materials used in the study.

Type	Brand	Code	Description	Composition	pH	Manufacturer
Material	Cerasmart 270	CS	Resin nanoceramic blocks	Inorganic phase (71 wt%): Silica nanoparticles, barium glass ceramic particles, glass phase containing strontium and aluminum, nanoceramic filler. Organic phase (29 wt%): UDMA, Bis-MEPP, DMA, other auxiliary dimethacrylates		GC Corp., Tokyo, Japan
	Vita Enamic	VE	Polymer-infiltrated ceramic network blocks	Inorganic phase (86 wt%): Feldspathic ceramic network (SiO_2_, Al_2_O_3_, Na_2_O, K_2_O), zirconia, sodium aluminosilicate glass. Organic phase (14 wt%): UDMA, TEGDMA, Bis-GMA, PMMA, and other dimethacrylates		VITA Zahnfabrik, Bad Säckingen, Germany
	Filtek Universal	FU	Nanohybrid RC for universal use	Inorganic fillers (76.5 wt%): Zirconia/silica nanoparticles, spherical silica nanoparticles (~20 nm), aggregate structures (0.6–1.4 µm). Organic resin matrix (23.5 wt%): Bis-GMA, UDMA, TEGDMA, Bis-EMA		3M ESPE, St. Paul, MN, USA
	everStick NET	ES	PMMA-based glass fiber strip	Fiber phase (44–46 vol%): E-glass fiber (7–10 µm). Polymer matrix (54–56 vol%): Bis-GMA, PMMA		GC Corp., Tokyo, Japan
Solution	Testonic Artificial Saliva	AS	Artificial saliva (ISO 7491:2000) [40]	Sodium, potassium, calcium, chloride, bicarbonate and phosphate ions, viscosity enhancers, preservatives	6.8	Colin Kimya, Istanbul, Turkey
	Kühne Apple Vinegar	ACV	Apple cider vinegar	Acetic acid, plant extract (apple), antioxidant	4.1	Carl Kühne KG, Hamburg, Germany
	Andorex Mouthwash	CXA	Chlorhexidine/alcohol mouthwash	0.12% chlorhexidine digluconate, benzydamine hydrochloride, Patent Blue V, glycerol, polysorbate 20, tartrazine (E102), ethanol, purified water	5.2	Humanis Health, Istanbul, Turkey
	Klorhex Plus Mouthwash	CX	Chlorhexidine/no alcohol mouthwash	0.12% chlorhexidine digluconate, flurbiprofen, Patent Blue V (E131), sorbitol, glycerol, purified water	5.5	Drogsan GmbH, Ankara, Turkey
	One Drop Only	HRA	Chlorhexidine/alcohol mouthwash	Aqua, menthol, thymol, eugenol, benzyl benzoate, alcohol, menthol, peppermint oil, tree resin, sage oil, tea tree oil, limonene, linalool, citral, water	6.8	One Drop Only GmbH, Berlin, Germany
	Agarta Mouthwash	HR	Chlorhexidine/no alcohol mouthwash	Bay leaf, licorice root, sage oil, laurel extract, chamomile flower extract, peppermint oil, green tea, propolis extract, menthol, glycerin, water	6.8	Agarta Cosmetics, Ankara, Turkey

UDMA: Urethane dimethacrylate; Bis-MEPP: Bisphenol A glycidyl methacrylate-phosphoric acid ester; DMA: Dimethylacrylamide; SiO_2_: Silicon dioxide; Al_2_O_3_: Aluminum oxide; Na_2_O: Sodium oxide; K_2_O: Potassium oxide; TEGDMA: Triethylene glycol dimethacrylate; Bis-GMA: Bisphenol A glycidyl methacrylate; PMMA: Polymethyl methacrylate; Bis-EMA: Bisphenol A ethoxylate dimethacrylate. Abbreviations used throughout text and figures.

**Table 2 polymers-17-02552-t002:** Results of the ANOVA.

	Source	Sum of Squares	df	F	*p* Value	Partial Eta Squared
Ra	Material	0.811	4	25.429	**<0.001**	0.086
	Solution	1.611	5	40.415	**<0.001**	0.158
	Time	4.737	3	198.009	**<0.001**	0.355
	Material-solution	0.053	20	0.333	0.998	0.006
	Material-time	0.142	12	1.488	0.122	0.016
	Solution-time	0.979	15	8.182	**<0.001**	0.102
	Material-solution-time	0.170	60	0.354	1.000	0.019
∆E_00_	Material	57.330	4	53.796	**<0.001**	0.210
	Solution	175.886	5	132.035	**<0.001**	0.449
	Time	119.740	2	224.717	**<0.001**	0.357
	Material-solution	18.050	20	3.387	**<0.001**	0.077
	Material-time	14.037	8	6.586	**<0.001**	0.061
	Solution-time	23.517	10	8.827	**<0.001**	0.098
	Material-solution-time	8.441	40	0.792	0.819	0.038
∆WI_D_	Material	8.067	4	6.505	**<0.001**	0.031
	Solution	91.938	5	59.304	**<0.001**	0.268
	Time	448.372	2	723.042	**<0.001**	0.641
	Material-solution	30.041	20	4.844	**<0.001**	0.107
	Material-time	16.662	8	6.717	**<0.001**	0.062
	Solution-time	93.496	10	30.154	**<0.001**	0.271
	Material-solution-time	16.392	40	1.322	0.090	0.061

Significant association (*p* < 0.05) indicated in bold. Ra: Surface roughness; ∆E_00_: Color change; ∆WI_D_: Whiteness Index change. Abbreviations used throughout text and figures.

**Table 3 polymers-17-02552-t003:** Surface roughness means (µm) ± standard deviations.

		AS	ACV	CXA	CX	HRA	HR	Total
FU	T0	0.18 ± 0.11	0.13 ± 0.05 ^a^	0.18 ± 0.11 ^a^	0.17 ± 0.07 ^a^	0.15 ± 0.03	0.19 ± 0.06	0.17 ± 0.08 ^a^
	T1	0.17 ± 0.04 ^A^	0.27 ± 0.10 ^Ba^	0.25 ± 0.23 ^Bb^	0.22 ± 0.04 ^ABa^	0.22 ± 0.09 ^AB^	0.20 ± 0.08 ^AB^	0.22 ± 0.12 ^b^
	T2	0.20 ± 0.05 ^A^	0.40 ± 0.10 ^Bb^	0.34 ± 0.15 ^Bc^	0.32 ± 0.09 ^Cb^	0.30 ± 0.07 ^C^	0.24 ± 0.05 ^A^	0.30 ± 0.11 ^c^
	T3	0.24 ± 0.06 ^A^	0.43 ± 0.08 ^Bc^	0.38 ± 0.17 ^Bc^	0.33 ± 0.10 ^Cb^	0.33 ± 0.08 ^C^	0.26 ± 0.06 ^A^	0.33 ± 0.11 ^d^
	Total	0.20 ± 0.07 ^A^	0.31 ± 0.14 ^B^	0.29 ± 0.18 ^C^	0.26 ± 0.10 ^C^	0.25 ± 0.10 ^D^	0.22 ± 0.07 ^E^	0.25 ± 0.12
FUS	T0	0.18 ± 0.06	0.18 ± 0.05 ^a^	0.16 ± 0.10 ^a^	0.17 ± 0.06 ^a^	0.18 ± 0.05 ^a^	0.18 ± 0.06 ^a^	0.17 ± 0.06 ^a^
	T1	0.18 ± 0.06 ^A^	0.28 ± 0.02 ^Bb^	0.25 ± 0.11 ^Ab^	0.22 ± 0.04 ^Ab^	0.23 ± 0.02 ^Aa^	0.19 ± 0.07 ^Aa^	0.22 ± 0.07 ^b^
	T2	0.21 ± 0.03 ^A^	0.40 ± 0.11 ^Bc^	0.35 ± 0.09 ^Bc^	0.32 ± 0.10 ^Bc^	0.28 ± 0.05 ^Ab^	0.26 ± 0.05 ^Ab^	0.30 ± 0.10 ^c^
	T3	0.24 ± 0.04 ^A^	0.45 ± 0.09 ^Bd^	0.37 ± 0.06 ^Cd^	0.36 ± 0.08 ^Cd^	0.31 ± 0.06 ^Ac^	0.29 ± 0.09 ^Ac^	0.34 ± 0.10 ^d^
	Total	0.20 ± 0.05 ^A^	0.33 ± 0.13 ^B^	0.28 ± 0.12 ^C^	0.27 ± 0.10 ^C^	0.25 ± 0.07 ^D^	0.23 ± 0.08 ^E^	0.26 ± 0.10
FUM	T0	0.18 ± 0.08	0.14 ± 0.04 ^a^	0.16 ± 0.04 ^a^	0.17 ± 0.04 ^a^	0.16 ± 0.06 ^a^	0.16 ± 0.09 ^a^	0.16 ± 0.06 ^a^
	T1	0.17 ± 0.03 ^A^	0.28 ± 0.04 ^Bb^	0.25 ± 0.09 ^Ab^	0.22 ± 0.04 ^Aa^	0.20 ± 0.06 ^Aa^	0.18 ± 0.06 ^Aa^	0.22 ± 0.06 ^b^
	T2	0.20 ± 0.06 ^A^	0.36 ± 0.11 ^Bc^	0.37 ± 0.09 ^Bc^	0.29 ± 0.07 ^Bb^	0.28 ± 0.16 ^Ab^	0.26 ± 0.13 ^Ab^	0.29 ± 0.12 ^c^
	T3	0.23 ± 0.03 ^A^	0.41 ± 0.07 ^Bd^	0.36 ± 0.07 ^Bc^	0.30 ± 0.11 ^Ac^	0.33 ± 0.13 ^Bc^	0.32 ± 0.12 ^Cc^	0.32 ± 0.11 ^d^
	Total	0.19 ± 0.06 ^A^	0.30 ± 0.12 ^B^	0.28 ± 0.11 ^C^	0.25 ± 0.09 ^C^	0.24 ± 0.12 ^D^	0.23 ± 0.12 ^E^	0.25 ± 0.11
CS	T0	0.21 ± 0.14	0.18 ± 0.11 ^a^	0.18 ± 0.05 ^a^	0.18 ± 0.06 ^a^	0.15 ± 0.03 ^a^	0.27 ± 0.39 ^a^	0.19 ± 0.17 ^a^
	T1	0.22 ± 0.05 ^A^	0.35 ± 0.03 ^Bb^	0.32 ± 0.04 ^Bb^	0.30 ± 0.01 ^Ab^	0.27 ± 0.05 ^Ab^	0.26 ± 0.02 ^Aa^	0.29 ± 0.05 ^b^
	T2	0.23 ± 0.03 ^A^	0.44 ± 0.08 ^Bc^	0.40 ± 0.09 ^Bc^	0.37 ± 0.04 ^Bc^	0.35 ± 0.10 ^Cc^	0.31 ± 0.08 ^Ca^	0.35 ± 0.10 ^c^
	T3	0.28 ± 0.05 ^A^	0.47 ± 0.06 ^Bd^	0.44 ± 0.10 ^Bd^	0.39 ± 0.05 ^Cd^	0.42 ± 0.14 ^Bd^	0.35 ± 0.08 ^Ab^	0.39 ± 0.10 ^d^
	Total	0.23 ± 0.08 ^A^	0.36 ± 0.13 ^B^	0.33 ± 0.12 ^C^	0.31 ± 0.09 ^C^	0.30 ± 0.13 ^C^	0.30 ± 0.20 ^D^	0.30 ± 0.14
VE	T0	0.16 ± 0.01	0.16 ± 0.05 ^a^	0.19 ± 0.12 ^a^	0.19 ± 0.13 ^a^	0.18 ± 0.03 ^a^	0.15 ± 0.01	0.17 ± 0.07 ^a^
	T1	0.16 ± 0.04	0.21 ± 0.04 ^a^	0.21 ± 0.06 ^a^	0.19 ± 0.06 ^a^	0.17 ± 0.02 ^a^	0.16 ± 0.02	0.18 ± 0.04 ^a^
	T2	0.17 ± 0.06 ^A^	0.35 ± 0.04 ^Bb^	0.31 ± 0.05 ^Bb^	0.26 ± 0.03 ^Ba^	0.25 ± 0.06 ^Bb^	0.21 ± 0.02 ^C^	0.26 ± 0.07 ^b^
	T3	0.22 ± 0.06 ^A^	0.37 ± 0.03 ^Bb^	0.34 ± 0.07 ^Bc^	0.30 ± 0.05 ^Bb^	0.29 ± 0.09 ^Cc^	0.23 ± 0.06 ^D^	0.29 ± 0.08 ^c^
	Total	0.18 ± 0.05 ^A^	0.27 ± 0.10 ^B^	0.26 ± 0.10 ^C^	0.23 ± 0.09 ^C^	0.22 ± 0.07 ^D^	0.19 ± 0.05 ^E^	0.22 ± 0.08
Total	T0	0.18 ± 0.09 ^a^	0.16 ± 0.07	0.17 ± 0.09 ^a^	0.17 ± 0.08	0.17 ± 0.04	0.19 ± 0.18 ^a^	0.17 ± 0.10 ^a^
	T1	0.18 ± 0.05 ^Ab^	0.28 ± 0.07 ^B^	0.25 ± 0.12 ^Ba^	0.23 ± 0.05 ^B^	0.22 ± 0.06 ^C^	0.20 ± 0.06 ^Bb^	0.23 ± 0.08 ^b^
	T2	0.20 ± 0.05 ^Ac^	0.39 ± 0.09 ^B^	0.35 ± 0.10 ^Bb^	0.31 ± 0.08 ^B^	0.29 ± 0.10 ^C^	0.26 ± 0.08 ^Da^	0.30 ± 0.10 ^c^
	T3	0.24 ± 0.05 ^Ab^	0.43 ± 0.07 ^B^	0.38 ± 0.10 ^Ba^	0.34 ± 0.09 ^B^	0.34 ± 0.11 ^C^	0.29 ± 0.09 ^Da^	0.33 ± 0.10 ^d^
	Total	0.20 ± 0.06 ^A^	0.31 ± 0.13 ^B^	0.29 ± 0.13 ^B^	0.26 ± 0.10 ^B^	0.25 ± 0.10 ^C^	0.23 ± 0.12 ^C^	0.26 ± 0.11

ANOVA *p* value: Means sharing the same superscript lowercase letter within a column are not significantly different (*p* > 0.05). Means sharing the same superscript uppercase letter within a row are not significantly different (*p* > 0.05). FU: Filtek Universal; FUS: Filtek Universal with superficial fiber strip; FUM: Filtek Universal with mid-layer fiber strip; CS: Cerasmart 270; VE: Vita Enamic. Abbreviations used throughout text and figures.

**Table 4 polymers-17-02552-t004:** Color change means ± standard deviations.

		AS	ACV	CXA	CX	HRA	HR	Total
FU	T0–T1	1.47 ± 0.26 ^Aa^	2.19 ± 0.83 ^Ba^	2.01 ± 0.49 ^Ba^	1.71 ± 0.55 ^Aa^	1.70 ± 0.53 ^Aa^	1.19 ± 0.65 ^Aa^	1.71 ± 0,64 ^a^
	T0–T2	1.95 ± 0.34 ^Ab^	2.68 ± 0.79 ^Bb^	2.51 ± 0.46 ^ABb^	2.19 ± 0.42 ^Bb^	1.99 ± 0.63 ^ABa^	1.56 ± 0.35 ^Ca^	2.15 ± 0.63 ^b^
	T2–T3	0.75 ± 0.38 ^Ac^	0.98 ± 0.23 ^ABc^	0.88 ± 0.55 ^ABc^	1.08 ± 0.15 ^ABc^	1.01 ± 0.21 ^ABb^	0.41 ± 0.14 ^Bb^	0.85 ± 0.37 ^c^
	Total	1.39 ± 0.59 ^A^	1.95 ± 0.98 ^B^	1.79 ± 0.84 ^BC^	1.66 ± 0.61 ^C^	1.56 ± 0.63 ^AC^	1.06 ± 0.64 ^D^	1.57 ± 0.78
FUS	T0–T1	0.37 ± 0.22 ^A^	1.69 ± 0.88 ^Ba^	1.73 ± 0.68 ^Ba^	1.23 ± 0.95 ^C^	0.32 ± 0.37 ^A^	0.25 ± 0.15 ^A^	0.93 ± 0.88 ^a^
	T0–T2	0.78 ± 0.47 ^A^	2.18 ± 1.02 ^Bb^	2.08 ± 0.66 ^Bb^	1.60 ± 0.65 ^C^	0.75 ± 0.38 ^A^	0.61 ± 0.36 ^A^	1.33 ± 089 ^b^
	T2–T3	0.37 ± 0.35 ^A^	1.05 ± 0.49 ^Bc^	1.15 ± 0.18 ^Bc^	0.92 ± 0.62 ^B^	0.33 ± 0.26 ^A^	0.28 ± 0.17 ^A^	0.68 ± 052 ^c^
	Total	0.51 ± 0.40 ^A^	1.64 ± 0.92 ^B^	1.65 ± 0.66 ^B^	1.25 ± 0.78 ^C^	0.47 ± 0.39 ^A^	0.38 ± 0.29 ^A^	0.98 ± 0.8
FUM	T0–T1	0.57 ± 0.31 ^Aa^	2.08 ± 0.93 ^Ba^	1.97 ± 0.66 ^Ba^	1.33 ± 0.60 ^Ca^	0.61 ± 0.46 ^Aa^	0.27 ± 0.14 ^A^	1.14 ± 0.9 ^a^
	T0–T2	1.06 ± 0.40 ^Ab^	2.40 ± 0.49 ^Ba^	2.34 ± 0.69 ^Ba^	1.79 ± 0.36 ^Cb^	0.91 ± 0.60 ^Ab^	0.64 ± 0.46 ^A^	1.52 ± 085 ^b^
	T2–T3	0.57 ± 0.43 ^Aa^	0.96 ± 0.32 ^Bb^	0.91 ± 0.67 ^Bb^	0.40 ± 0.14 ^Bc^	0.36 ± 0.17 ^Ba^	0.27 ± 0.10 ^B^	0.58 ± 0.44 ^c^
	Total	0.73 ± 0.44 ^A^	1.81 ± 0.88 ^B^	1.74 ± 0.89 ^B^	1.17 ± 0.71 ^C^	0.63 ± 0.49 ^A^	0.39 ± 0.33 ^A^	1.08 ± 0.85
CS	T0–T1	0.65 ± 0.31 ^Aa^	1.82 ± 0.57 ^Ba^	1.74 ± 0.96 ^Ba^	1.45 ± 1.16 ^Ca^	0.75 ± 0.32 ^Aa^	0.67 ± 0.40 ^Aa^	1.18 ± 0.84 ^a^
	T0–T2	1.10 ± 0.31 ^Aa^	2.16 ± 0.54 ^Ba^	2.06 ± 0.65 ^Ba^	1.89 ± 1.48 ^Cb^	1.19 ± 0.28 ^Ab^	1.16 ± 0.91 ^Ab^	1.59 ± 090 ^b^
	T2–T3	0.40 ± 0.18 ^Ab^	0.90 ± 0.24 ^Bb^	0.69 ± 0.59 ^Ab^	0.40 ± 0.32 ^Bc^	0.40 ± 0.19 ^Ba^	0.46 ± 0.40 ^Ab^	0.54 ± 0.38 ^c^
	Total	0.72 ± 0.39 ^A^	1.63 ± 0.71 ^B^	1.49 ± 0.9 ^B^	1.25 ± 1.24 ^C^	0.78 ± 0.42 ^A^	0.76 ± 0.67 ^A^	1.10 ± 0.86
VE	T0–T1	0.19 ± 0.11 ^A^	1.63 ± 0.42 ^Ba^	0.61 ± 0.54 ^Ba^	1.23 ± 0.36 ^Ca^	0.22 ± 0.12 ^A^	0.18 ± 0.10 ^A^	0.84 ± 073 ^a^
	T0–T2	0.44 ± 0.23 ^A^	1.25 ± 0.43 ^Ba^	2.01 ± 0.29 ^Cb^	1.79 ± 0.39 ^Db^	0.49 ± 0.42 ^A^	0.27 ± 0.24 ^A^	1.04 ± 0.76 ^b^
	T2–T3	0.20 ± 0.11 ^A^	0.94 ± 0.33 ^Bb^	0.81 ± 0.31 ^Bc^	0.50 ± 0.26 ^Ac^	0.54 ± 0.29 ^A^	0.26 ± 0.12 ^B^	0.53 ± 036 ^c^
	Total	0.27 ± 0.19 ^A^	1.27 ± 0.48 ^B^	1.47 ± 0.63 ^C^	1.17 ± 0.63 ^C^	0.42 ± 0.32 ^A^	0.24 ± 0.17 ^A^	0.81 ± 0.67
Total	T0–T1	0.65 ± 0.51 ^A^	1.88 ± 0.75 ^B^	1.81 ± 0.68 ^B^	1.39 ± 0.77 ^C^	0.72 ± 0.65 ^A^	0.51 ± 0.31 ^A^	1.16 ± 085 ^a^
	T0–T2	1.07 ± 0.61 ^A^	2.13 ± 0.82 ^B^	2.20 ± 0.58 ^B^	1.85 ± 0.78 ^C^	1.06 ± 0.69 ^A^	0.85 ± 0.68 ^A^	1.53 ± 0.88 ^b^
	T2–T3	0.46 ± 0.35 ^A^	0.97 ± 0.32 ^B^	0.89 ± 0.50 ^B^	0.66 ± 0.43 ^B^	0.52 ± 0.33 ^B^	0.34 ± 0.22 ^B^	0.64 ± 0.43 ^c^
	Total	0.72 ± 0.56 ^A^	1.66 ± 0.83 ^B^	1.63 ± 0.80 ^B^	1.30 ± 0.83 ^C^	0.77 ± 0.62 ^D^	0.57 ± 035 ^E^	1.11 ± 0.83

ANOVA *p* value: Means sharing the same superscript lowercase letter within a column are not significantly different (*p* > 0.05). Means sharing the same superscript uppercase letter within a row are not significantly different (*p* > 0.05). FU: Filtek Universal; FUS: Filtek Universal with superficial fiber strip; FUM: Filtek Universal with mid-layer fiber strip; CS: Cerasmart 270; VE: Vita Enamic. Abbreviations used throughout text and figures.

**Table 5 polymers-17-02552-t005:** Whiteness index change means ± standard deviations.

		AS	ACV	CXA	CX	HRA	HR	Total
FU	T0–T1	−1.24 ± 0.28 ^Aa^	−1.8 ± 0.66 ^Ba^	−1.62 ± 0.57 ^Ba^	−1.45 ± 0.69 ^Ba^	−1.31 ± 0.67 ^Ba^	0.26 ± 0.14 ^C^	−1.20 ± 0.86 ^a^
	T0–T2	−1.60 ± 0.53 ^Aa^	−2.38 ± 1.02 ^Bb^	−2.17 ± 0.83 ^Cb^	−1.82 ± 0.80 ^Da^	−1.33 ± 0.58 ^Ea^	0.51 ± 0.28 ^E^	−1.46 ± 1.18 ^b^
	T2–T3	0.51 ± 0.28 ^b^	0.69 ± 0.27 ^c^	0.56 ± 0.37 ^c^	0.72 ± 0.23 ^b^	0.55 ± 0.32 ^b^	0.69 ± 0.27	0.62 ± 0.29 ^c^
	Total	−0.77 ± 1.01 ^A^	−1.18 ± 1.53 ^AB^	−1.07 ± 1.34 ^AB^	−0.85 ± 1.29 ^BC^	−0.69 ± 1.04 ^BC^	0.48 ± 0.29 ^C^	−0.68 ± 1.26
FUS	T0–T1	−0.31 ± 0.27 ^Aa^	−1.31 ± 0.94 ^Ba^	−1.44 ± 0.47 ^Ca^	−1.08 ± 0.84 ^Aa^	−0.25 ± 0.40 ^Aa^	−0.21 ± 0.23 ^Aa^	−0.77 ± 0.77 ^a^
	T0–T2	−0.67 ± 0.53 ^Aa^	−1.85 ± 0.82 ^Bb^	−1.79 ± 0.96 ^Ba^	−1.22 ± 0.75 ^Aa^	−0.47 ± 0.42 ^Aa^	−0.54 ± 0.51 ^Aa^	−1.09 ± 0.87 ^b^
	T2–T3	0.24 ± 0.20 ^Ab^	0.44 ± 0.26 ^Ac^	0.79 ± 0.30 ^Bb^	0.60 ± 0.46 ^Ab^	0.22 ± 0.24 ^Bb^	0.17 ± 0.13 ^Bb^	0.41 ± 0.36 ^c^
	Total	−0.24 ± 0.52 ^A^	−0.90 ± 1.22 ^B^	−0.81 ± 1.32 ^B^	−0.56 ± 1.08 ^B^	−0.17 ± 0.46 ^C^	−0.19 ± 0.44 ^C^	−0.48 ± 0.95
FUM	T0–T1	−0.33 ± 0.42 ^Aa^	−1.59 ± 1.01 ^Ba^	−1.59 ± 0.63 ^Ba^	−1.15 ± 0.71 ^Ba^	−0.47 ± 0.41 ^Aa^	−0.10 ± 0.14 ^Aa^	−0.87 ± 0.74 ^a^
	T0–T2	−0.92 ± 0.37 ^Ab^	−2.11 ± 0.38 ^Bb^	−2.05 ± 0.84 ^Ba^	−1.51 ± 0.44 ^Ca^	−0.51 ± 0.45 ^Aa^	−0.42 ± 0.34 ^Aa^	−1.25 ± 0.84 ^b^
	T2–T3	0.29 ± 0.25 ^Ac^	0.66 ± 0.31 ^Ac^	0.62 ± 0.31 ^Ab^	0.41 ± 0.30 ^Ab^	0.13 ± 0.09 ^Bb^	0.23 ± 0.12 ^Ab^	0.39 ± 0.31 ^c^
	Total	−0.31 ± 0.61 ^A^	−1.02 ± 1.37 ^B^	−1.01 ± 1.33 ^B^	−0.75 ± 0.98 ^B^	−0.28 ± 0.47 ^C^	−0.09 ± 0.35 ^C^	−0.58 ± 1.01
CS	T0–T1	−0.36 ± 0.32 ^Aa^	−1.29 ± 1.01 ^Ba^	−1.21 ± 1.68 ^Ba^	−1.14 ± 0.77 ^Ba^	−0.34 ± 0.33 ^Aa^	−0.59 ± 0.56 ^Aa^	−0.82 ± 0.96 ^a^
	T0–T2	−0.88 ± 0.28 ^Ab^	−1.75 ± 0.54 ^Ba^	−1.88 ± 0.79 ^Cb^	−1.59 ± 1.43 ^Ba^	−0.78 ± 0.43 ^Aa^	−0.83 ± 0.77 ^Aa^	−1.28 ± 0.88 ^b^
	T2–T3	0.25 ± 0.11 ^c^	0.52 ± 0.26 ^b^	0.60 ± 0.47 ^c^	0.52 ± 0.32 ^b^	0.23 ± 0.16 ^b^	0.36 ± 0.31 ^b^	0.42 ± 0.32 ^c^
	Total	−0.33 ± 0.53 ^A^	−0.83 ± 1.19 ^B^	−0.82 ± 1.49 ^B^	−0.73 ± 1.31 ^B^	−0.29 ± 0.52 ^C^	−0.35 ± 077 ^C^	−0.56 ± 1.05
VE	T0–T1	0.02 ± 0.12 ^Aa^	−1.04 ± 0.47 ^Ba^	−1.30 ± 0.80 ^Ca^	−1.10 ± 0.72 ^Ba^	−0.08 ± 0.12 ^Aa^	−0.30 ± 0.35 ^A^	−0.63 ± 0.72 ^a^
	T0–T2	−0.40 ± 0.17 ^Aa^	−1.15 ± 0.42 ^Ba^	−1.47 ± 0.63 ^Ca^	−1.54 ± 0.62 ^Ca^	−0.45 ± 0.33 ^Aa^	−0.33 ± 0.54 ^A^	−0.89 ± 0.69 ^b^
	T2–T3	0.18 ± 0.11 ^b^	0.47 ± 0.32 ^b^	0.59 ± 0.31 ^b^	0.24 ± 0.22 ^b^	0.40 ± 0.33 ^b^	0.08 ± 0.17	0.32 ± 0.27 ^c^
	Total	−0.06 ± 0.28 ^A^	−0.57 ± 0.85 ^B^	−0.72 ± 1.12 ^B^	−0.84 ± 0.94 ^B^	−0.04 ± 0.44 ^C^	−0.18 ± 0.41 ^C^	−0.40 ± 0.80
Total	T0–T1	−0.44 ± 0.51 ^Aa^	−1.41 ± 0.86 ^Ba^	−1.43 ± 0.91 ^Ba^	−1.18 ± 0.73 ^Ca^	−0.49 ± 0.59 ^Aa^	−0.19 ± 0.42 ^Da^	−0.86 ± 0.75 ^a^
	T0–T2	−0.89 ± 0.55 ^Ab^	−1.85 ± 0.77 ^Bb^	−1.87 ± 0.80 ^Cb^	−1.54 ± 0.86 ^Cb^	−0.71 ± 0.55 ^Ab^	−0.32 ± 0.67 ^Da^	−1.20 ± 0.92 ^b^
	T2–T3	0.29 ± 0.23 ^Ac^	0.56 ± 0.29 ^Ac^	0.63 ± 0.35 B^c^	0.50 ± 0.35 ^Ac^	0.31 ± 0.28 ^Ac^	0.30 ± 0.29 ^Bb^	0.43 ± 0.33 ^c^
	Total	−0.34 ± 0.67 ^A^	−0.90 ± 1.25 ^B^	−0.89 ± 1.32 ^B^	−0.74 ± 1.12 ^C^	−0.29 ± 0.66 ^A^	−0.06 ± 0.56 ^D^	−0.54 ± 1.03

ANOVA *p* value: Means sharing the same superscript lowercase letter within a column are not significantly different (*p* > 0.05). Means sharing the same superscript uppercase letter within a row are not significantly different (*p* > 0.05). FU: Filtek Universal; FUS: Filtek Universal with superficial fiber strip; FUM: Filtek Universal with mid-layer fiber strip; CS: Cerasmart 270; VE: Vita Enamic. Abbreviations used throughout text and figures.

## Data Availability

The original contributions presented in this study are included in the article. Further inquiries can be directed to the corresponding author.

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
