# Peer review of "Effects of Apple Vinegar, Mouthwashes, and Bleaching on Color Stability and Surface Properties of Fiber-Reinforced and Non-Reinforced Restorative Materials"

_polymers, 2025, doi:10.3390/polym17182552_

Round 1

Reviewer 1 Report

Comments and Suggestions for Authors

Reviewer comments

The manuscript deals with an important topic of “Effects of Apple Vinegar, Mouthwashes, and Bleaching on Color Stability and Surface Properties of Fiber-Reinforced and Non-Reinforced Restorative Materials”. The aim of this study was to investigate the effects of apple cider vinegar (ACV), various mouthwashes and bleaching on the color and surface roughness of fiber strip-reinforced and unreinforced restorative materials. However, certain limitations exist that, if addressed, could enhance the comprehensiveness and applicability of the findings. It is suggested to be accepted after major revision. The specific comments are the following.

  1. The study uses an accelerated aging model (T1 = 2 years, T2 = 5 years) but lacks justification for the chosen time intervals (1 day and 2.5 days). Suggestion: Cite specific literature validating this conversion (e.g., ISO 13356 or studies correlating immersion time with clinical aging). Alternatively, provide a brief calculation (e.g., based on daily mouthwash exposure duration).
  2. To facilitate the reviewers' accurate identification of the locations for revision, please mark the line numbers on your paper.
  3. The sample size calculation (n = 300) is mentioned, but the effect size (0.25) and power (0.95) are not contextualized for the study’s specific outcomes (Ra, ΔE₀₀, ΔWI_D). Suggestion: Clarify how the effect size was derived (e.g., from pilot data or prior studies on similar materials). Specify whether it aligns with thresholds for clinically significant changes in roughness/color (e.g., ΔE₀₀ > 1.6 for perceptibility).
  4. Ra values exceeded the bacterial adhesion threshold (0.2 µm), but the clinical implications are not discussed. Suggestion: Add a paragraph in the Discussion comparing the observed Ra values (e.g., 0.33 µm for ACV at T3) to established thresholds for biofilm accumulation or secondary caries risk.
  5. SEM results are mentioned but not described or interpreted (e.g., no images or qualitative observations). Suggestion: Include representative SEM images (5000×–25,000×) in a supplementary file and describe key findings (e.g., pitting in ACV groups vs. cracking in CXA). Relate morphology changes to Ra/ΔE₀₀ data. Location: Methods/Results?
  6. The section of introduction is weak. For the part of “Introduction”, the lack of literature references lowers the credibility and scientific nature of the research basis. A large sum of references is necessary to be added. It is suggested to establish a quantitative relationship between preparation technology and design development. Compared with other works, such as Journal of Building Engineering, 2025: 113207 (https://doi.org/10.1016/j.jobe.2025.113207).
  7. The study notes that mid-layer fiber strips (FUM) affected color more than superficial strips (FUS) but does not explain why. Suggestion: Discuss potential mechanisms (e.g., light scattering differences due to fiber depth or interfacial degradation) and cite studies on fiber-matrix interactions.
  8. Consistently use abbreviations after first definition. Consider adding a footnote to Table 1: "Abbreviations used throughout text and figures." Location: Table 1 caption and figures.
  9. The language description of the article needs to be standardized. Please check it carefully and avoid using unprofessional words and sentences.

Author Response

Reviewer Comment:

The manuscript addresses an important topic on the effects of apple vinegar, mouthwashes, and bleaching on the color stability and surface properties of fiber-reinforced and non-reinforced restorative materials. However, there are some limitations that could enhance the comprehensiveness and applicability of the findings. The manuscript is suggested for acceptance after a major revision.

Response:

We sincerely thank the reviewer for recognizing the importance and relevance of our research topic. We fully acknowledge the reviewer’s constructive feedback regarding the need to address the study’s limitations and to improve the comprehensiveness of the findings. In the revised version, we carefully considered these remarks and made substantial revisions throughout the manuscript. Specifically, we expanded the Introduction with additional background and references, clarified methodological details, strengthened the presentation and interpretation of results, and improved the clarity of tables and figures.

Reviewer Comment 1:

The study employed an accelerated aging model (T1 = 2 years, T2 = 5 years); however, the chosen time intervals (1 day and 2.5 days) are not justified. It is suggested to reference specific literature validating this conversion (e.g., ISO 13356 or studies correlating immersion times with clinical aging). Alternatively, a brief calculation could be provided (e.g., based on daily mouthwash exposure duration).  

Response 1:

We appreciate the reviewer’s valuable suggestion. In the revised manuscript, we clarified the rationale for the accelerated aging model. Specifically, we explained that mouth rinses are typically used for about 2 minutes per day, which corresponds to approximately 12 h of total annual exposure. Therefore, 12 h of continuous immersion was considered equivalent to 1 year of clinical use, as reported in previous studies. Based on this calculation, immersion periods of 24 h and 60 h (2.5 days) were applied to simulate 2 years (T1) and 5 years (T2) of clinical exposure, respectively. This explanation and the relevant references have been added to the Materials and Methods section. (Lines 243–249)

Reviewer Comment 2:

The sample size calculation (n = 300) is mentioned, but the effect size (0.25) and power (0.95) are not contextualized for the study’s specific outcomes (Ra, ΔE₀₀, ΔWI_D). Suggestion: Clarify how the effect size was derived (e.g., from pilot data or prior studies on similar materials). Specify whether it aligns with thresholds for clinically significant changes in roughness/color (e.g., ΔE₀₀ > 1.6 for perceptibility).

Response 2:

We appreciate the reviewer’s comment. A medium effect size (f = 0.25; η² ≈ 0.06) was selected based on statistical conventions and previous CAD/CAM restorative studies. Clinically relevant thresholds for Ra, ΔE₀₀, and ΔWID were also considered, resulting in a total sample size of n = 300 with a power of 0.95 and α = 0.05. This explanation has been added to the Materials and Methods section. (Lines 184–190)

Reviewer Comment 3:

Ra values exceeded the bacterial adhesion threshold (0.2 µm), but the clinical implications are not discussed. Suggestion: Add a paragraph in the Discussion comparing the observed Ra values (e.g., 0.33 µm for ACV at T3) to established thresholds for biofilm accumulation or secondary caries risk.

Response 3:

We thank the reviewer for this valuable comment. In the revised version, we have added a new paragraph in the Discussion section that directly compares the observed Ra values of different groups with the established thresholds and discusses their clinical implications for biofilm accumulation, secondary caries, and periodontal risk.

To avoid redundancy and ensure a coherent flow, we also revised the surrounding paragraphs:

The introductory paragraph on Ra, which was previously general, was shortened to present only the null hypothesis and the threshold definitions.

The new paragraph, inserted immediately afterwards, provides specific Ra values with explicit comparison to clinical thresholds, as suggested by the reviewer.

The paragraph discussing the study by VaizoÄŸlu et al. was retained for literature- and mechanism-based discussion, but numerical values already covered in the new paragraph were removed to prevent repetition. (Lines 524–540, 581–587)

Reviewer Comment 4:

SEM results are mentioned but not described or interpreted (e.g., no images or qualitative observations). Suggestion: Include representative SEM images (5000×–25,000×) in a supplementary file and describe key findings (e.g., pitting in ACV groups vs. cracking in CXA). Relate morphology changes to Ra/ΔE₀₀ data. Location: Methods/Results.

Response 4:

We thank the reviewer for this valuable comment. In the revised manuscript, we have addressed this point in three ways:

Supplementary Figures: Representative SEM images at 5000×, 10,000×, and 25,000× magnification for each experimental group have been added to the Supplementary File. These images illustrate typical morphological features such as pitting, microcracking, filler–matrix debonding, and polymer phase loss under different storage conditions.

Results Section: A detailed qualitative description of the SEM findings has been incorporated into the Results section. These descriptions summarize the key morphological changes at different magnification levels. (xxx)

Discussion Section: The relationship between these SEM observations and surface roughness and color change values has been elaborated in the Discussion section. (Lines 424–440, 447–464, 466–480, 482–497, 498–516, 715–759)

Reviewer Comment 5:

The section of introduction is weak. For the part of “Introduction”, the lack of literature references lowers the credibility and scientific nature of the research basis. A large sum of references is necessary to be added. It is suggested to establish a quantitative relationship between preparation technology and design development. Compared with other works, such as Journal of Building Engineering, 2025: 113207 (https://doi.org/10.1016/j.jobe.2025.113207).

Response 5:

We thank the reviewer for this valuable observation. In the revised manuscript, the Introduction section has been significantly strengthened with the addition of multiple up-to-date references to enhance the scientific foundation of the study. Specifically:

Additional studies were cited regarding the nanostructural and optical properties of universal RCs.

The section on acidic challenges was expanded to include apple cider vinegar (ACV), highlighting its effects on dentin, enamel, and the gap in evidence for CAD/CAM restorative materials.

The discussion on mouthwashes was broadened to differentiate alcohol-containing and alcohol-free formulations and to highlight the potential discoloration effects of herbal products.

The part on bleaching agents was elaborated with references to the oxidative effects of conventional systems and recent developments in controlled-release systems (e.g., HPS).

For fiber-reinforced RCs, additional literature was included to emphasize the lack of studies addressing the effect of fiber strip location on optical and surface properties.

We have added a recent reference (q19) on contemporary color measurement methods to strengthen the scientific basis of the introduction.

Furthermore, a dedicated paragraph was added to establish a quantitative link between preparation technology (CAD/CAM fabrication, fiber strip positioning, surface conditioning) and design development, as recommended by the reviewer. (Lines 44–40, 51–58, 77–131, 137–144)

Reviewer Comment 6:

The study notes that mid-layer fiber strips (FUM) affected color more than superficial strips (FUS) but does not explain why. Suggestion: Discuss potential mechanisms (e.g., light scattering differences due to fiber depth or interfacial degradation) and cite studies on fiber-matrix interactions.

Response 6:

We thank the reviewer for this valuable comment. In the revised manuscript, we have expanded the Discussion section to address possible mechanisms underlying the higher color change observed in the FUM group compared to the FUS group. Specifically, we noted that fibers located in the mid-layer may increase light scattering within the bulk material, thereby enhancing color differences, whereas superficially placed fibers may partially reflect and diffuse light, resulting in an optical masking effect. Furthermore, interfacial degradation processes, such as water sorption–induced microcracks, may be more pronounced when fibers are embedded deeper in the resin matrix, which could exacerbate discoloration over time. We have also added references supporting the role of fiber–matrix interactions and refractive index matching in optical behavior. (Lines 677–701, 709–714)

Reviewer Comment 7:

Consistently use abbreviations after first definition. Consider adding a footnote to Table 1: "Abbreviations used throughout text and figures." Location: Table 1 caption and figures.

Response 7:

We thank the reviewer for this valuable suggestion. In accordance with the comment, the note “Abbreviations used throughout text and figures” has been added to the footnote of Table 1 and to the captions of all relevant figures. To ensure consistency and clarity for the readers, the same note has also been included under Tables 2–4, where abbreviations are used. In addition, we have verified whether the abbreviations were properly defined at their first occurrence. (Lines 183,217,230,328,352,375,405,802,803).

Reviewer Comment 8:

The language description of the article needs to be standardized. Please check it carefully and avoid using unprofessional words and sentences.

Response 8:

We thank the reviewer for this valuable comment. The manuscript has been carefully revised, the language standardized, and unprofessional expressions removed to ensure clarity and an appropriate academic tone. All corrections have been highlighted in red in the revised manuscript.

Reviewer 2 Report

Comments and Suggestions for Authors

The manuscript addresses an important clinical topic and is generally well designed, with clear methodology and robust statistical analysis. The findings are relevant, but some points require clarification before acceptance.

Major Points for Clarification:

  1. Immersion times: The authors state that T1 (1 day) simulates 2 years and T2 (2.5 days) simulates 5 years. Why was this accelerated aging model chosen, and what is the justification for this equivalence?
  2. Bleaching effects: Despite using 25% HP, bleaching showed minimal influence compared to ACV. Explain the reason why bleaching did not significantly alter surface properties.
  3. Fiber strip position: Why did the strip position affect color/whiteness but not roughness? A mechanistic explanation would strengthen the discussion.
  4. Clinical relevance: Since roughness values exceeded the bacterial adhesion threshold, how should clinicians interpret the balance between acceptable color stability and increased risk of plaque retention?

Author Response

Reviewer Comment

The manuscript addresses an important clinical topic and is generally well designed, with clear methodology and robust statistical analysis. The findings are relevant, but some points require clarification before acceptance.

Response

The manuscript addresses an important clinical topic and is generally well designed, with clear methodology and robust statistical analysis. The findings are relevant, but some points require clarification before acceptance.

Reviewer Comment 1:

Immersion times: The authors state that T1 (1 day) simulates 2 years and T2 (2.5 days) simulates 5 years. Why was this accelerated aging model chosen, and what is the justification for this equivalence?

Response 1:

We thank the reviewer for this insightful comment. The accelerated aging model applied in our study was based on the adaptation of daily mouthwash exposure (approximately 2 minutes per day) into laboratory conditions. In the literature, 12 h of continuous immersion has been reported to correspond to approximately 1 year of clinical use. Accordingly, 24 h of immersion represents about 2 years (T1), and 60 h (2.5 days) corresponds to about 5 years (T2) of clinical service. This model is consistent with previous reports that applied similar accelerated immersion protocols. To address the reviewer’s concern, this explanation has now been explicitly added to the Methods section of the revised manuscript. (Lines 243–247)

Reviewer Comment 2:

Bleaching effects: Despite using 25% HP, bleaching showed minimal influence compared to ACV. Explain the reason why bleaching did not significantly alter surface properties.

Response 2:

We thank the reviewer for this valuable comment. The controlled radical release of HPS, together with the thermogelation properties of its carrier system, preserved surface integrity and resulted in only minimal alterations. Moreover, the short-term exposure of office-type bleaching limited degradation compared with continuous immersion in acidic solutions. This explanation has been added to the revised Discussion section. (Lines 590–598)

Reviewer Comment 3:

Fiber strip position: Why did the strip position affect color/whiteness but not roughness? A mechanistic explanation would strengthen the discussion.

Response 3:

We thank the reviewer for this comment. The influence of fiber position on color and whiteness was attributed to light scattering and optical masking within the material. (xxx)

In contrast, surface roughness was primarily determined by matrix–filler interactions under chemical exposure, and fiber position therefore had no decisive effect on this parameter. This mechanistic explanation has been added to the revised Discussion section. (Lines 690–708)

Reviewer Comment 4:

Clinical relevance: Since roughness values exceeded the bacterial adhesion threshold, how should clinicians interpret the balance between acceptable color stability and increased risk of plaque retention?

Response 4:

We thank the reviewer for this comment. Although Ra values in some groups exceeded the 0.2 µm bacterial adhesion threshold, none of them surpassed the 0.5 µm tongue perceptibility threshold. Thus, the surfaces remained clinically tolerable; however, the potential risk of biofilm formation and plaque retention should not be overlooked. Accordingly, even when color stability is clinically acceptable, increased roughness warrants patient guidance and regular professional maintenance. This clarification has been added to the revised Discussion section. (Lines 529–545)

Round 2

Reviewer 1 Report

Comments and Suggestions for Authors

All the reviewers' comments have been addressed. The manuscript can be considered for acceptance.